# Effects of traditional Chinese medicine in the treatment of patients with central serous chorioretinopathy: A systematic review and meta-analysis

**Shuting Ru[1,2☯], Jian Sun[3☯], Wanyu Zhou[1], Dong Wei[1], Hang Shi[1], Yu Liang[1], Jianguo Wu[1], Wu Sun[1]\*, Liqun Chu[1]\***

1 Department of Ophthalmology, Xiyuan Hospital of China Academy of Chinese Medical Sciences, Beijing, China, 2 Chinese Academy of Chinese Medical Sciences, Beijing, China, 3 Department of Ophthalmology, Shanghai Pudong New Area Zhoupu Hospital, Shanghai, China

☯ These authors contributed equally to this work.

\* wsss55555@126.com (WS); chuliqunok@126.com (LC)

**Data Availability Statement:** All relevant data are within the manuscript and its Supporting Information files.

## Abstract

Several studies have reported the efficacy of traditional Chinese medicine (TCM) for central serous chorioretinopathy (CSC), while some ophthalmologists are concerned that TCM may be a risk factor for CSC as some chinese herbs contain hormonal ingredients. This study aimed to evaluate the efficacy and safety of TCM in treating patients with CSC. Randomized controlled trials (RCTs) and observational studies of TCM for CSC were searched up to July 10, 2023 on the following biological databases without language and publication time restrictions: PubMed, Ovid Medline, Embase, Cochrane Library, The Chinese National Knowledge Infrastructure Database (CNKI), Technology Periodical Database (VIP), Wanfang, and Chinese Biomedical Literature Service System (SinoMed). Review Manager V.5.4.1 and Stata 14 software were used for data analysis. Finally, thirty-eight studies were finally included including 23 RCTs and 15 cohort studies. The meta-analysis showed that compared with the routine treatment alone, the combination of TCM can not only reduce the recurrence rate (OR = 0.29, 95% CI: 0.21,0.40; $I^2$ = 0%) and central retinal thickness (CRT) (MD = - 35.63, 95% CI: - 45.96,-25.30; $I^2$ = 89%) of CSC, but improve patients' best corrected visual acuity (BCVA) (SMD = 0.86, 95% CI: 0.62,1.11; $I^2$ = 77%); additionally, it has no obvious side effects compared with routine treatment (OR = 0.72, 95% CI: 0.39,1.34; $I^2$ = 10%). Overall, this study shows that the use of TCM does not increase the risk of CSC recurrence; on the contrary, the combination of TCM may reduce the recurrence of CSC and improve BCVA and CRT in patients with CSC compared with conventional treatment.

## Introduction

Central serous chorioretinopathy (CSC) is a chorioretinal disease characterized by serous detachment of the neurosensory retina, accompanied by retinal pigment epithelium (RPE)

**Funding:** This work was supported by "Beijing Natural Science Foundation" (No. 7244501), "Major Research Project of Acupuncture and Moxibustion Clinical Discipline, Science and Technology Innovation Project, Chinese Academy of Chinese Medical Sciences" (No. C12021A03513) and "Independent Selection Project of Chinese Academy of Chinese Medical Sciences" (No. ZZ16-XRZ-024). There was no additional external funding received for this study. The funders had no role in study design, data collection and analysis, decision to publish, or preparation of the manuscript.

**Competing interests:** The authors have declared that no competing interests exist.

lesions and hyperpermeability of the choroid [1]. Typical manifestations of CSC include loss of central vision, central blindness, microvision, or deformity. It is one of the 10 most common disorders in the back of the eye and is a common cause of mild to moderate vision impairment. The reported incidence of CSC is 10 cases per 100,000 men and 2 cases per 100,000 women [2]. In most cases, CSC is self-limiting, and patients' vision and symptoms recover within 3 months [2]. However, according to published literature [3], the recurrence rate of CSC is 25–50%, which can lead to poor prognosis and even permanent blindness [1].

Currently known risk factors for CSC include genetic risk [4], corticosteroids [5], endocrinological abnormalities [1, 6], pregnancy [7], etc. Corticosteroids, in particular, are considered to be strongly associated with CSC and are recommended not to be used in the treatment of CSC because of the risk of further exacerbating the disease [8].

Current treatments for CSC mainly include photodynamic therapy (PDT), laser therapy, and anti-vascular endothelial growth factor (anti-VEGF) therapy [9]. However, PDT is not available in all countries and PDT itself has serious side effects, including ocular events such as choroidal ischemia and subsequent retinal atrophy, as well as systemic events such as headache, back pain, nausea, dyspnea, dizziness, and syncope [9–11]. Conventional laser photocoagulation is only suitable for treating extrafoveal leakage points, and its efficacy against CSC remains unclear. In addition, adverse events such as scotoma, vision loss, reduced contrast sensitivity, and/or macular neovascularization (MNV) may occur in the treated area due to damage to the neuralretina-RPE-Bruch's membrane [9, 12]. The efficacy of anti-VEGF treatment in treating CSC still lacks clear evidence and the treatment is limited to patients with concurrent macular neovascularization [9, 13]. Therefore, it is necessary to explore new adjuvant therapeutic measures for CSC.

Traditional Chinese medicine (TCM), a characteristic medical treatment in China, is widely used in China and even Asia. TCM has been widely used in a variety of fundus diseases involving edema and hemorrhagic lesions, including neovascular age-related macular degeneration, diabetic retinopathy, and retinal vein obstruction, and is believed to promote the absorption of fundus edema [14–17]. Therefore, TCM is also commonly used to treat CSC in China. However, some clinicians are concerned that Chinese medicines are risk factors for CSC because some of them contain hormonal components or hormone-like effects [18, 19]. Currently, there is a lack of conclusive evidence to prove the relationship between TCM and CSC. Therefore, we conducted a meta-analysis of recent studies on herbal interventions in patients with CSC, especially those containing hormonal components or having hormone-like effects, to observe the effects of TCM on CSC.

## Methods

### Study registration and ethics statements

This meta-analysis was registered with the International Prospective Register of Systematic Reviews (PROSPERO; registration number: CRD42023428288) and strictly adhered to the Preferred Reporting Items for Systematic Reviews and Meta-analyses (PRISMA) [20]. As this study was a reanalysis of published papers and did not involve additional human trials, it did not require ethics committee approval or consent.

### Inclusion criteria

This meta-analysis included Randomized controlled trials (RCTs) or observational studies involving patients diagnosed with CSC. Interventions that contained TCM in the intervention group were included, including oral administration of herbs, herbal medicines, herbal capsules, and proprietary Chinese medicines. Studies in the intervention group that combine

TCM treatment with treatment in the control group were also included, while studies that combine treatment outside of TCM were excluded. In addition, studies of intravenous input and topical application of herbal medicines were excluded. Interventions in the control group contained conventional treatment measures such as medications to improve fundus microcirculation, vitamin-based supplements, PDT, anti-VEGF, and laser therapy, as well as placebo and no treatment. The recurrence rate of CSC and best corrected visual acuity (BCVA) were set as primary outcomes, and central retinal thickness (CRT) and adverse events were set as secondary outcomes.

## Exclusion criteria

Studies involving any of the following were not included: 1) studies containing herbal medicines in both control and intervention groups; 2) the treatment in the intervention group included treatments out of TCM in addition to the control group interventions; 3) case series; 4) no full text; and 5) studies in which key information was unclear or unknown and no results were available after contacting the authors.

## Search strategy

Relevant literature was searched in the following databases: PubMed, Ovid Medline, Embase, Cochrane Library, The Chinese National Knowledge Infrastructure Database, Technology Periodical Database (VIP), Wanfang, and Chinese Biomedical Literature Service System (SinoMed). The search time was from inception to July 10, 2023, without language and publication time restrictions. In addition, relevant web pages were also manually searched (www. clinicaltrials.gov; www.clinicaltrialsregister.eu; trialsearch.who.int) for ongoing trials or unpublished clinical trial reports. The specific search strategy can be found in S1 Table.

## Data extraction

Two reviewers conducted a literature search independently (JS and SR). After screening out the duplicate documents in EndNote software, a preliminary review was carried out by reading the titles and abstracts of the retrieved documents. The literature that satisfied the inclusion and exclusion criteria was read in full to determine its eligibility for further inclusion. For eligible trials, two reviewers (LY and JW) independently extracted information based on a predesigned standardized template, including (1) study characteristics (study year, country, and study type); (2) patient characteristics, including the information of sample size, sex, age, CSC type (acute, < 6 months; chronic, ≥ 6 months), etc; (3) details of intervention measures (TCM composition, frequency, and duration of treatment); and (4) clinical outcome indicators. Any differences between the two reviewers were resolved through communication and negotiation with an arbiter.

## Quality assessment

The methodological quality of the included studies was evaluated independently by two reviewers (JS and HS) according to the Cochrane risk-of-bias tool for randomized trials (RoB 2.0) as follows [21]: randomization process, deviations from the intended interventions, missing outcome data, measurement of the outcome and selection of the reported result. For each item, we divided the research into "high", "unclear", and "low" risk of bias. The overall risk of bias for each study was evaluated based on S2 Table. The quality of observational studies was evaluated using the Newcastle - Ottawa quality assessment scale (NOS) [22]. When there was insufficient information to make a judgment, we inquired about relevant information to the

corresponding author via email. Any controversies were settled through consultation with the third reviewer (WS).

### Data analysis

ReviewManager (RevMan) version 5.4.1 (The Cochrane Collaboration, Oxford, UK) was used for the meta-analysis. Continuous outcome variables were calculated by mean differences (MDs) or standard mean differences (SMDs) with 95% confidence intervals (CIs), and dichotomous outcome variables were calculated by combined odds ratio (ORs) with 95% CIs. When the heterogeneity of outcome variables was low ($P > 0.10$, $I^2 < 50\%$), the fixed-effect model was used; otherwise, the random-effect model was used. Publication bias was evaluated visually by creating funnel plots *via* RevMan 5.4.1 version, as well as by conducting Egger's regression test using STATA 14.0 version (Stata Corp, College Station, TX, USA) [23]. Subgroup analysis was performed by intervention type (with or without TCM contained hormone component), CSC type (acute, < 6 months; chronic, ≥ 6 months), intervention type (PDT, laser, et al), or intervention course. Sensitivity analyses were performed to observe changes in synthetic results according to the following operations: 1) excluding low-quality studies; 2) excluding studies with small sample size; 3) excluding studies with the largest sample size; 4) excluding studies containing Chinese patent medicine; 5) or switching between fixed and random effects models. For indicators that were not sufficient for the meta-analysis, a narrative description was made [24].

### Quality of evidence

The quality of the pooled evidence for all the outcomes was judged by two independent reviewers with extensive work experience as ophthalmologists (LC) and TCM practitioners (WZ) according to the Grading of Recommendations Assessment, Development, and Evaluation (GRADE) system [25]. The strength of evidence was graded as "high", "moderate", "low" or "very low" based on five assessment items: risk of bias, inconsistency, indirectness, imprecision, and other considerations. Any controversies were settled through consultation with the third reviewer (WS).

## Results

### Literature search

A total of 1826 articles were included, of which 988 studies were removed due to duplication. After reading the titles and abstracts, 360 articles remained (478 articles were removed, including 167 irrelevant articles, 189 reviews, and 122 case reports and case series). Of these, 114 used inappropriate controls, 128 lacked indicators of interest, 71 lacked key information, and 9 involved duplicate publications. Finally, 38 studies were included [26–63] (Fig 1). See S1 Table for details.

### Characteristics of the included studies

All studies were conducted in China and included 23 RCTs [26, 27, 29–42, 44,45, 50, 53–56], 14 retrospective cohort studies [43, 46–49, 51, 52, 57–63], and 1 prospective cohort study [28]. A total of 2849 patients (3063 eyes) were included. Among them, 23 studies (1739 individuals 1891 eyes) involving acute CSC (disease duration < 6 months), 6 studies (421 cases 442 eyes) of chronic CSC, and 9 studies (689 individuals 730 eyes) of mixed type. By reviewing the available information, the age range of the patients was 16–60 years, and the duration of the disease was 2 days-10 years. The interventions in the TCM group consisted of 35 articles on herbal medicines

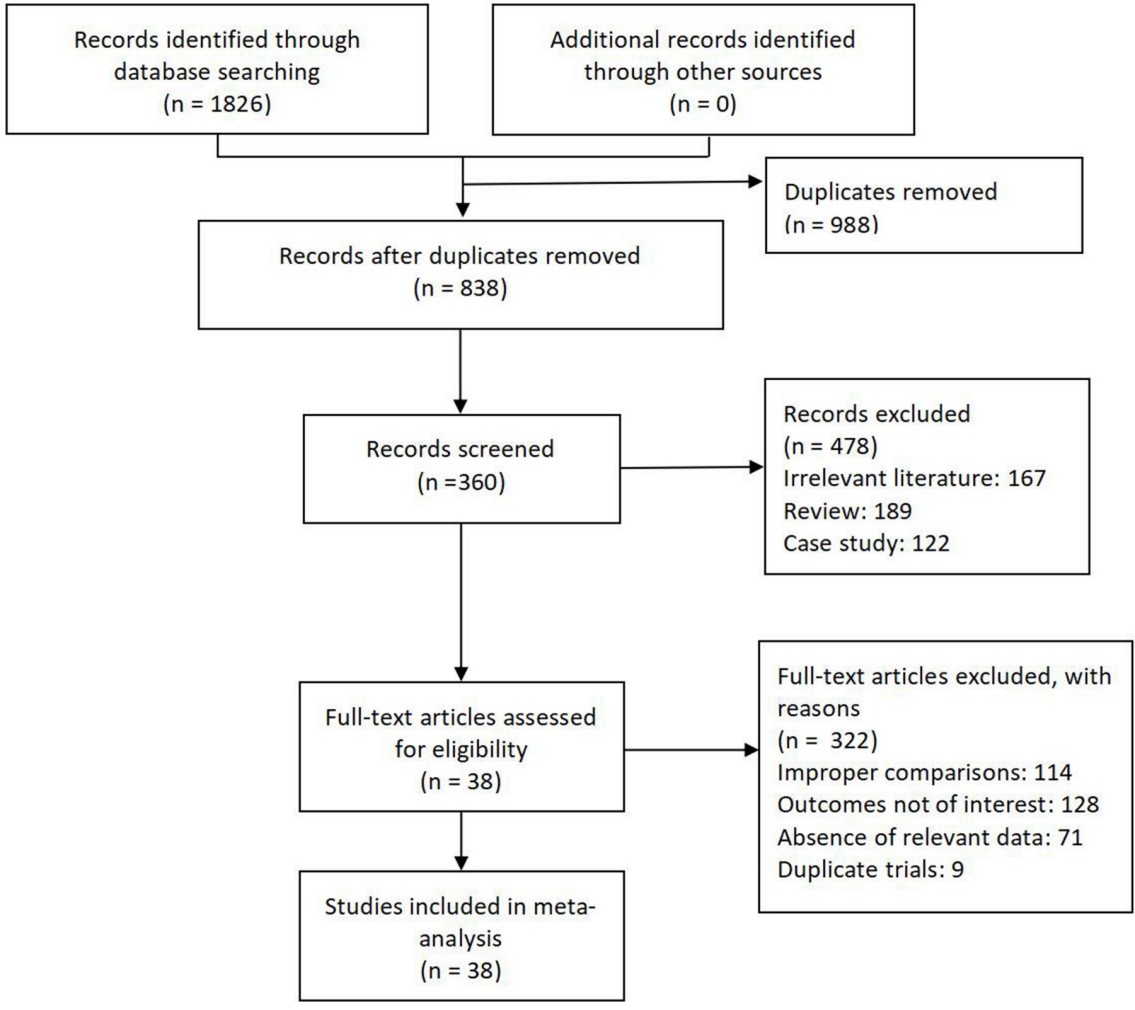

**Fig 1. Study flow diagram.**

and 3 articles on proprietary Chinese medicines. According to pharmacological studies, herbal medicines that contain corticosteroid components or have hormone-like effects include antler velvet [64], ginseng [65], radix astragali [66], angelica sinensis [67], wolfberry fruit [68], schizandra [69], kudzu vine root, licorice, rhubarb, tragacanth, and herba epimedii [70].

This paper included 24 studies in which herbal medicines contained hormonal components or hormone-like effects [27, 30–32, 34–36, 39–47, 50, 51, 56–59, 61, 62].

Interventions in the control group included 30 articles on vitamin supplements/retinal microcirculation-improving drugs, 7 articles on laser, 2 articles on PDT, and 1 article on no treatment. The follow-up period ranged from 3 months to 2 years.

Regarding outcome indicators, 24 studies mentioned recurrence rate, 23 mentioned BCVA, 15 mentioned CRT, 7 mentioned adverse events, and 1 mentioned SRF area. Of these, BCVA was recorded in 18 studies using a standard visual acuity chart (decimal) [26–28, 30–35, 37, 39, 40, 41, 44, 48, 50, 53, 54], 2 using a 5 m Standard Logarithmic Visual Acuity (5SL) [42, 43], and 3 using the logarithm of the minimum angle of resolution (logMAR) [36, 38, 47]. Table 1 lists the specific information from the studies included.

**Table 1. The basic information of included studies.**

| Author (Country) | Study Design | Cases (E/C) | Age (years) (range, mean ± sd) | Sex (M/F) | State of Disease | Intervention | Duration | follow-up period | Outcomes |
|---|---|---|---|---|---|---|---|---|---|
| He 2023 [26] | RCT | 41 (41 eyes)/ 41 (41 eyes) | E: 44.2±6.9 C: 43.6±7.4 | E: 32/ 9 C: 41/ 10 | E: (10.1 ±2.3) d C: (10.6 ±2.8) d | E: TCM (poria, grifola umbellata, rhizoma alismatis, largehead atractylodes rhizome, cassia twig)+C C: calcium dobesilate capsules | 8 w | NR | ①②③ |
| Liu 2022 [27] | RCT | 38 (38 eyes)/ 38 (38 eyes) | E: 39.57 ±6.52 C:39.89 ±6.71 | E: 26/ 12 C: 24/ 14 | E: (23.08 ±5.84) d C: (23.37 ±5.87) d | E: TCM (concha haliotidis, rhizoma alismatis, chinese yam, poria, grifola umbellata, cassia twig, salvia miltiorrhiza, codonopsis, largehead atractylodes rhizome, dried rehmannia root, plantain seed, herba lycopi, **licorice**)+C C: calcium dobesilate capsules | 3 m | NR | ①② |
| Shang 2022 [28] | Prospective Cohort study | 18 (18 eyes)/ 20 (20 eyes) | E: 42.83 ±5.43 C: 43.85 ±7.87 | E: 16/ 2 C: 17/ 3 | NR | E: TCM (poria, grifola umbellata, rhizoma alismatis, largehead atractylodes rhizome, cassia twig)+C C: laser therapy | 1 m | 6 m | ①②③④ |
| Li N 2021 [29] | RCT | 30 (30 eyes)/ 30 (30 eyes) | E: 36.45 ±3.62 C: 36.58 ±3.63 | E: 18/ 12 C: 20/ 10 | E: (2.78 ±0.35) m C: (2.82 ±0.37) m | E: TCM (poria, grifola umbellata, rhizoma alismatis, largehead atractylodes rhizome, cassia twig) C: vitamin B1+vitamin C+calcium dobesilate capsules | 1 m | NR | |
| Xu 2021 [30] | RCT | 22 (22 eyes)/ 22 (22 eyes) | E: 44.73 ±8.20 C: 42.55 ±9.88 | E: 19/ 3 C: 18/ 4 | E: (7–30) d C: (7–28) d | E: TCM (poria, rhizoma alismatis, grifola umbellata, plantain seed, codonopsis, largehead atractylodes rhizome, semen coicis, orange peel, **angelica sinensis**, radix paeoniae rubra, radix achyranthes, **herba epimedii**, fenugreek)+C C: Laser therapy+vitamin B1+mecobalamin | 2 m | NR | ①② |
| Sha 2021 [31] | RCT | 40 (40 eyes)/ 40 (40 eyes) | E: 35.60 ±2.57 C: 34.71 ±2.87 | E: 19/ 21 C: 20/ 20 | E: (8.32 ±1.26) m C: (7.85 ±1.32) m | E: TCM (poria, grifola umbellata, rhizoma alismatis, largehead atractylodes rhizome, cassia twig, **angelica sinensis**, radix paeoniae alba, prepared rehmannia root, ligusticum wallichii) +C C: calcium dobesilate capsules | 6 w | NR | ①② |
| LI JX 2021 [32] | RCT | 20 (20 eyes)/ 20 (20 eyes) | E: 35.60 ±2.57 C: 34.71 ±2.87 | E: 12/ 8 C: 11/ 9 | E: (7.15 ±0.23) m C: (7.12 ±0.22) m | E: TCM (plantain seed, codonopsis, semen coicis, radix puerariae, salvia miltiorrhiza, curcuma aromatica, cortex moutan, largehead atractylodes rhizome, poria, orange peel, rhizoma corydalis, nutgrass galingale rhizome, **licorice**)+C C: troxerutin+inosine tablet+21 Super-Vita | NR | NR | ① |
| Bi 2020 [33] | RCT | 51 (51 eyes)/ 52 (52 eyes) | E: 36.54 ±3.22 C: 37.43 ±2.81 | E: 38/ 13 C: 33/ 19 | E: (10.5 ±1.33) d C: (13.28 ±1.69) d | E: TCM (poria, grifola umbellata, rhizoma alismatis, largehead atractylodes rhizome, cassia twig)+C C: anisodine injection | 1 m | NR | ①③ |
| Pang 2018 [34] | RCT | 37 (37 eyes)/ 37 (37 eyes) | E: 39.58 ±8.54 C:39.64 ±8.46 | E: 30/ 7 C: 28/ 9 | E: (0.98 ±0.25) y C: (0.99 ±0.28) y | E: TCM(poria, grifola umbellata, rhizoma alismatis, largehead atractylodes rhizome, cassia twig, **licorice**, dried rehmannia root, concha haliotidis, codonopsis, herba lycopi, salvia miltiorrhiza, chinese yam, plantain seed)+C C: vitamin B1+inosine tablets+adenosine disodium triphosphate tablets | 3 m | 1 y | ①②④ |
| Cao 2018 [35] | RCT | 31 (31 eyes)/ 30 (30 eyes) | E: 43.63 ±4.76 C: 42.65 ±5.27 | E: 26/ 5 C: 25/ 5 | E: (11.3 ±7.4) m C: (11.6 ±6.9) m | E: TCM (largehead atractylodes rhizome, mangnolia officinalis, papaya, banksia rose, amomum tsao-ko, pericarpium arecae, poria, rhizoma zingiberis, **honey-fried licorice root**, fresh ginger, fructus ziziphi jujubae) C: mecobalamin | 6 w | NR | ①② |

(*Continued*)

**Table 1.** (Continued)

| Author (Country) | Study Design | Cases (E/C) | Age (years) (range, mean ± sd) | Sex (M/F) | State of Disease | Intervention | Duration | follow-up period | Outcomes |
|---|---|---|---|---|---|---|---|---|---|
| Sun 2017 [36] | RCT | 40 (40 eyes)/ 34 (34 eyes) | E: 47.30 ±6.77 C: 46.15 ±5.75 | E: 33/ 7 C: 28/ 8 | Acute CSC | E: TCM (poria, largehead atractylodes rhizome, **radix astragali**, grifola umbellata, rhizoma alismatis, leonurus, semen benincasae, cassia twig, radix bupleuri, motherwort fruit, **licorice**) C: vitamin B complex+vitamin C+inosine tablets | 6 w | NR | ①② |
| Li 2017 [37] | RCT | 42 (42 eyes)/ 41 (41 eyes) | E: 36.05 ±13.24 C: 35.91 ±13.16 | E: 37/ 5 C: 37/ 4 | E: (3.03 ±1.26) m C: (3.11 ±1.12) m | E: complex thrombolysis caps + C C: vitamin B1+vitamin C+vitamin E | 2 m | 3 m | ①②④ |
| Luo 2017 [38] | RCT | 24 (24 eyes)/ 24 (24 eyes) | E: 38.21 ±8.65 C: 37.38 ±8.33 | E: 37/ 5 C: 37/ 4 | Acute CSC | E: TCM (largehead atractylodes rhizome, poria, grifola umbellata, rhizoma alismatis) C: No treatment | 1m | 3 m | ①②④ |
| Kuang 2017 [39] | RCT | 42 (47 eyes)/ 42 (46 eyes) | E: 37.5±5.5 C: 37.8±5.9 | E: 36/ 6 C: 35/ 7 | E: (0.5 ±0.1) y C: (0.6 ±0.1) y | E: TCM (peach seed, dried rehmannia root, carthamus tinctorius L, fructus aurantii, radix achyranthes, radix paeoniae rubra, **angelica sinensis**, platycodon grandiflorus, ligusticum wallichii, radix bupleuri, **licorice**)+C C: vitamin B1+vitamin B6+anisodine injection | 1 m | NR | ① |
| Zhang M 2017 [40] | RCT | 30 (30 eyes)/ 30 (30 eyes) | E: 25–48 C: 30–45 | E: 18/ 12 C:16/ 14 | E: (3–15) d C: (3–15) d | E: TCM(codonopsis, largehead atractylodes rhizome,orange peel, semen coicis, rhizoma alismatis, plantain seed, grifola umbellata, poria, ligusticum wallichii, salvia miltiorrhiza, seaweed, ecklonia kurome okam, **licorice**)+C C: vitamin B1+inosine tablets+bendazol | 1 m | NR | ①② |
| Xu 2017 [41] | RCT | 30 (30 eyes)/ 30 (30 eyes) | E: 39.13 ±7.42 C: 37.33 ±7.97 | E: 27/ 3 C:25/ 5 | E: (19.4 ±13.5) d C: (16.93 ±12.52) d | E: TCM (poria, largehead atractylodes rhizome, grifola umbellata, rhizoma alismatis, cassia twig, herba lycopi, codonopsis, chinese yam, salvia miltiorrhiza, plantain seed, dried rehmannia root, **licorice**, concha haliotidis) C: vitamin B1+vitamin C | 3 m | NR | ① |
| Zhu 2017 [42] | RCT | 25 (28 eyes)/ 22 (23 eyes) | E: 40.59 ±5.29 C: 40.52 ±5.94 | E: 22/ 3 C: 20/ 2 | E: (2.95 ±1.11) m C: (3.07 ±1.08) m | E: TCM (rhizoma alismatis, poria, grifola umbellata, cassia twig, radix bupleuri, **angelica sinensis**, radix paeoniae alba, largehead atractylodes rhizome, fresh ginger, mint, **honey-fried licorice root**)+C C: anisodine injection | 3 m | NR | ①②③ |
| Zhang RX 2017 [43] | Retrospective Cohort study | 35 (35 eyes)/ 35 (35 eyes) | E: 32.3±3.2 C: 30.5±4.2 | E: 31/ 4 C: 29/ 6 | E: (14.6 ±6.8) d C: (12.3 ±5.7) d | E: TCM (poria, cassia twig, largehead atractylodes rhizome, **licorice, radix astragali**, herba lycopi, grifola umbellata plantain seed, fritillaria cirrhosa, pinellia ternata, orange peel)+C C: laser therapy | 1 m | 1 y | ①②④ |
| Zhao 2015 [44] | RCT | 26 (26 eyes)/ 25 (25 eyes) | E: 39±8 C: 39±8 | E: 26/ 5 C: 25/ 4 | NR | E: TCM (codonopsis, radix paeoniae alba, **angelica sinensis**, poria, fructus psoraleae, **herba epimedii**, plantain seed, semen coicis)+C C: vitamin B1+vitamin C+ATP tablets | 2 m | 1 y | ①④ |
| Li 2014 [45] | RCT | 49 (49 eyes)/ 48 (48 eyes) | E: 23–56 C: 25–54 | E: 40/ 9 C: 42/ 6 | E: (10.22 ±5.39) d C: (11.40 ±5.16) d | E: TCM (**angelica sinensis**, radix paeoniae alba, radix bupleuri, poria, largehead atractylodes rhizome, **honey-fried licorice root**, mint, fresh ginger) C: vitamin B1+inosine tablets+mecobalamin | 2 m | 2 y | |

(*Continued*)

**Table 1.** (Continued)

| Author (Country) | Study Design | Cases (E/C) | Age (years) (range, mean ± sd) | Sex (M/F) | State of Disease | Intervention | Duration | follow-up period | Outcomes |
|---|---|---|---|---|---|---|---|---|---|
| Ou 2014 [46] | Retrospective Cohort study | 41 (44 eyes)/ 41 (44 eyes) | 20–51 | E: 28/ 13 C: 30/ 11 | NR | E: TCM (radix bupleuri, **angelica sinensis**, radix paeoniae alba, poria, largehead atractylodes rhizome, plantain seed, grifola umbellata, rhizoma alismatis, cassia twig, **honey-fried licorice root**)+C C: aescuven forte+lutein | 3 m | 1 y | |
| Ning 2014 [47] | Retrospective Cohort study | 20 (20 eyes)/ 20 (20 eyes) | E: 32.15 ±8.63 C: 34.07 ±9.01 | E: 18/ 2 C: 17/ 3 | E: (8.45 ±3.03) d C: (7.8 ±2.95) d | E: TCM (codonopsis, chinese yam, largehead atractylodes rhizome, poria, semen coicis, fructus amomi, rhizoma alismatis, platycodon grandiflorum, crude pollen typhae, eclipta alba, cortex moutan, ligusticum wallichii, radix bupleuri, curcuma aromatica salisb, **licorice**) C: vitamin B complex+vitamin C+ATP tablets +bendazol | 14 d | 1 y | ①④ |
| Zhang 2014 [48] | Retrospective Cohort study | 42 (47 eyes)/ 40 (45 eyes) | E: 38.9±4.5 C: 39.1±4.7 | E: 30/ 12 C: 28/ 12 | E: (0.6 ±0.1) y C: (0.8 ±0.2) y | E: TCM(poria, grifola umbellata, rhizoma alismatis, largehead atractylodes rhizome, cassia twig)+C C: laser therapy | 6 w | NR | ①② |
| Tang 2013 [49] | Retrospective Cohort study | 93 (102 eyes)/79 (88 eyes) | E: 43.91 ±8.07 C: 43.91±6.7 | E: 70/ 23 C: 58/ 21 | E: (8.93 ±3.87) d C: (9.25 ±3.6) d | E: complex thrombolysis caps + C C: iodized lecithin | 1 m | 1 y | |
| Chen 2013 [50] | RCT | 30 (32 eyes)/30 (31 eyes) | E: 30.13 ±6.74 C: 31.57 ±5.62 | E: 23/ 7 C: 21/ 9 | E: (8.10 ±2.70) d C: (6.70 ±2.97) d | E: TCM (largehead atractylodes rhizome, **radix astragali**, aprieot seed, cardomon, semen coicis, poria grifola umbellata, salvia miltiorrhiza, radix bupleuri)+C C: vitamin B complex +compound rutin tablets +ATP tablets | 3 w | 2 y | ①④ |
| Ju 2013 [51] | Retrospective Cohort study | 45 (45 eyes)/40 (40 eyes) | E: 26–51 C: 24–50 | E: 35/ 10 C: 32/ 8 | NR | E: TCM (semen coicis, plantain seed, hyacinth bean, poria, motherwort fruit, chinese dodder seed, chinese yam, succvinum amber, **radix astragali**)+C C: vitamin B+bendazol+ATP tablets | 2 m | 1 y | |
| Liang 2012 [52] | Retrospective Cohort study | 28 (28 eyes)/ 24 (24 eyes) | E: 32±2 C: 30.5±2.9 | E: 30/ 12 C: 28/ 12 | E: 2 d-10 m C: 5 d-12 m | E: TCM(poria, grifola umbellata, rhizoma alismatis, largehead atractylodes rhizome, cassia twig) C: Compound vitamin B tablets+ATP tablets +Inosine tablets+iodized lecithin tablets | 2 m | 1 y | |
| Xiang 2008 [53] | RCT | 28 (30 eyes)/ 29 (30 eyes) | E: 38.56 ±8.12 C: 36.63 ±8.57 | E: 21/ 7 C: 22/ 7 | E: (15.57 ±34.24) w C: (12.53 ±27.48) w | E: TCM(poria, grifola umbellata, rhizoma alismatis, largehead atractylodes rhizome, cassia twig)+C C: laser therapy | 6 w | 1 y | ①④ |
| Lin 2007 [54] | RCT | 25 (27 eyes)/ 23 (24 eyes) | E: 40.8±7.5 C: 39.1±7.3 | E: 17/ 8 C: 16/ 7 | E: (15.5 ±36.8) w C: (14.8 ±37.3) w | E: TCM(poria, grifola umbellata, rhizoma alismatis, largehead atractylodes rhizome, cassia twig)+C C: laser therapy | 6 w | 1 y | ①④ |
| Chen 2009 [55] | RCT | 60 (68 eyes)/ 60 (65 eyes) | 23–57 | NR | 3 d-1 m | E: TCM (poria, grifola umbellata, rhizoma alismatis, largehead atractylodes rhizome, cassia twig, aprieot seed, talc, ricepaperplant pith, cardomon, bamboo leaf, mangnolia officinalis, semen coicis, pinellia ternata) C: Compound vitamin B tablets+ATP tablets +Inosine tablets+compound rutin tablets | 6 w | 1 y | |

(*Continued*)

**Table 1.** (Continued)

| Author (Country) | Study Design | Cases (E/C) | Age (years) (range, mean ± sd) | Sex (M/F) | State of Disease | Intervention | Duration | follow-up period | Outcomes |
|---|---|---|---|---|---|---|---|---|---|
| Zhang 2009 [56] | RCT | 42 (47 eyes)/ 42 (46 eyes) | E: 38.4±12.3 C: 36.9±10.9 | E: 34/ 8 C: 35/ 7 | E: (5–60) d C: (5–55) d | E: TCM (codonopsis, largehead atractylodes rhizome, poria, semen coicis, plantain seed, orange peel, curcuma aromatica salisb, salvia miltiorrhiza, cortex moutan, **kudzu vine root**, corydalis tuber, rhizoma cyperi, **licorice**)+C C: vitamin E+ATP tablets+inosine tablets +compound rutin tablets+cobamamide | 45 d | 1 y | |
| Tang 2009 [57] | Retrospective Cohort study | 41 (50 eyes)/ 35 (43 eyes) | E: 30–50 (40.5) C: 28–48 (38.2) | E: 35/ 6 C: 32/ 3 | E: 7 d-10 y C: 5 d-10 y | E: TCM (peach seed, carthamus tinctorius L, ligusticum wallichii, **angelica sinensis**, fructus aurantii, curcuma aromatica salisb, grassleaf sweelflag rhizome, rhizoma alismatis, poria, plantain seed, pale butterflybush flower, **wolfberry fruit**, ligustrum lucidum ait, concha haliotidis, **licorice**)+C C: laser therapy | 30 d | 1 y | |
| Liu 2008 [58] | Retrospective Cohort study | 63 (70 eyes)/ 39 (43 eyes) | E: 15–55 (44.6) C: 18–53 (43.4) | E: 58/ 5 C: 33/ 6 | E: 30 d-10 y C: 7 d-10 y | E: TCM (**radix astragali**, radix paeoniae rubra, poria, dendrobium, salvia miltiorrhiza, **wolfberry fruit**, **angelica sinensis** lumbricus, grassleaf sweelflag rhizome, carthamus tinctorius l, ligusticum wallichii, radix, bupleuriradix rehmanniae, prepared radix rehmanniae)+C C: vitamin B1+vitamin C+ATP tablets+inosine tablets +venoruton | NR | 1 y | ④ |
| Li 2007 [59] | Retrospective Cohort study | 30 (33 eyes)/ 30 (33 eyes) | E: 15–55 (44.6) C: 18–53 (43.4) | 46/14 | (3–30) d | E: TCM (**wolfberry fruit**, chrysanthemum, prepared rehmannia root, cornus officinalis, chinese yam, cortex moutan, rhizoma alismatis, poria)+C C: vitamin B12+ATP tablets+aminopeptide iodide | 30 d | 6 m | ④ |
| Du 2003 [60] | Retrospective Cohort study | 42 (46 eyes)/ 37 (40 eyes) | E: 28–47 C: 27–50 | E: 28/ 14 C: 24/ 13 | E: 7 d-4 y C: 3 d-3 y | E: TCM (aprieot seed, talc, ricepaperplant pith, cardomon, bamboo leaf, mangnolia officinalis, semen coicis, pinellia ternata)+C C: inosine tablets+ATP tablets+venoruton | 2 m | 2 y | ④ |
| Shen 2003 [61] | Retrospective Cohort study | 64 (64 eyes)/ 60 (60 eyes) | E: 28–47 C: 27–50 | E: 44/ 20 C: 40/ 20 | NR | E: TCM (radix bupleuri, ligusticum wallichii, radix paeoniae alba, poria,largehead atractylodes rhizome, dried rehmannia root, chinese yam, cornus officinalis, rhizoma alismatis, cortex moutan, **schizandra**)+C C: vitamin B1/B12+vitamin C+nicotinic acid +inosine tablets+troxerutin | 2 m | 1 y | ④ |
| Yu 2002 [62] | Retrospective Cohort study | 100 (100 eyes)/ 86 (86 eyes) | E: 38.4±12.3 C: 36.9±10.9 | E: 88/ 12 C: 68/ 18 | E: (6.8 ±3.2) d C: (6.5 ±2.9) d | E: TCM (dried rehmannia root, prepared rehmannia root, radix asparagi, ophiopogon japonicus, dendrobe, rhizoma polygonati odorati, rhizoma alismatis, poria, plantain seed, salvia miltiorrhiza, sappan wood, ligustrum lucidum ait, **wolfberry fruit**, mulberry, **licorice**)+C C: vitamin B1+troxerutin | NR | 1 y | ④ |
| Zhang 2001 [63] | Retrospective Cohort study | 43 (45 eyes)/ 45 (48 eyes) | 16–60 | E: 33/ 10 C: 35/ 10 | (3–45) d | E: TCM (codonopsis, largehead atractylodes rhizome,orange peel, semen coicis, rhizoma alismatis, plantain seed, grifola umbellata, poria, ligusticum wallichii, salvia miltiorrhiza)+C C: vitamin B1+ATP tablets+iodized lecithin +bendazol+venoruton | NR | 6 m | ④ |

C, control; CSC, Central serous chorioretinopathy; D, days; M, months; NR: not reported; TCM, traditional chinese medicine; W, weeks; Y, years; ① best corrected visual acuity, BCVA; ② central retinal thickness, CRT; ③ adverse event; ④ recurrence rate. Herbal ingredients (bolded) indicate the presence of hormones or hormone-like effects.

## Risk of bias assessment

ROB was used to evaluate the risk of bias in 23 RCTs [26, 27, 29–42, 44, 45, 50, 53–56]. Eighteen studies reported randomization methods, 13 of which used random number tables [26, 30–37, 40–42, 50], two used lottery methods [29, 56], and three were based on the order of attendance [53–55]. No placebo was used in any of the studies. All studies did not mention allocation concealment, blinding of subjects, and outcome evaluation, and the risk of bias was determined to be "unclear". There was no case shedding in any of the studies. Ultimately, the overall risk of bias in 5 studies was assessed as "High" and 18 studies as "medium".

Fifteen observational studies were evaluated using the NOS [28, 43, 46–49, 51, 52, 57–63], and all the patients were hospital-confirmed. One study took a prospective cohort [28], and the method of randomization was unknown. Fourteen studies used a retrospective cohort study. All studies proposed controls for age and gender factors. Follow-up was implemented in all but one study [48]. Of these, 11 [43, 46, 47, 49, 51, 52, 57, 58, 60–62] performed follow-ups of sufficient duration (follow-up ≥1 year). Overall, 12 studies were rated as high quality (NOS ≥7). See Table 2 for details.

## Outcome measurements

**Recurrence rate.** Twenty-four studies mentioned the recurrence rate [28, 34, 37, 38, 43–47, 49–63], and the results suggested that the TCM group could reduce the recurrence rate of CSC compared with the conventional treatment group (OR = 0.29, 95% CI: 0.21,0.40; $I^2$ = 0%) (Fig 2). In addition, subgroup analyses showed that TCM reduced the recurrence rate regardless of whether it contained hormonal components or not.

**BCVA.** BCVA was recorded in 18 studies using a standard visual acuity chart (decimal) [26–28, 30–35, 37, 39, 40, 41, 44, 48, 50, 53, 54] and 2 using a 5SL chart [42, 43], with very high heterogeneity in the results ($I^2$ = 77%). The random-effect model indicated that there was a significant difference between the TCM group and the control group in terms of BCVA (SMD = 0.86, 95% CI: 0.62,1.11) (Fig 3A). In addition, subgroup analyses showed that the TCM group containing hormonal components also elevated BCVA in CSC patients compared to the conventional treatment group (SMD = 1.02, 95% CI: 0.65,1.38).

BCVA was recorded in 3 studies using the logarithm of the minimum angle of resolution (logMAR) [36, 38, 47], and the random-effects model showed no statistically significant difference between the TCM group and the conventional treatment group (MD = -0.02, 95% CI: -0.06,0.02) (Fig 3B).

**CRT.** CRT was mentioned in 15 studies [26–31, 34–38, 40, 42, 43, 48], with high heterogeneity in the results ($I^2$ = 89%). The random-effect model suggested that the TCM group had reduced CRT compared with that in the control group (MD = - 35.63, 95% CI: - 45.96,-25.30) (Fig 4). In addition, subgroup analyses showed that TCM reduced the recurrence rate regardless of whether it contained hormonal components (MD = - 38.47, 95% CI: - 53.86,-23.08) or not (MD = - 33.72, 95% CI: - 49.47,-17.97).

**Adverse events.** Adverse events were reported in only 6 studies [26, 28, 29, 33, 42, 49], mainly including nausea, gastrointestinal reactions, panic, and subcutaneous hemorrhage and hardening accompanied by subcutaneous injection of drugs. The random effects model showed that there was no statistically significant difference in the incidence of adverse events between the TCM group and conventional treatment group (OR = 0.72, 95% CI: 0.39,1.34; $I^2$ = 10%), regardless of whether the TCM contained hormonal components or not. (Fig 5).

**Table 2. Quality assessment of included studies.**

**Quality assessment of randomized controlled trials (Cochrane Risk of Bias tool)**

| Author | Sequence Generation | Allocation Concealment | Blinding of Participants, Personnel | Blinding of Outcome Assessors | Incomplete Outcome Data | Selective Outcome Reporting | Other Sources of Bias | Overall risk of bias |
|---|---|---|---|---|---|---|---|---|
| He et al.2023 [26] | Low risk | Unclear | High risk | Unclear | Low risk | Low risk | Low risk | Medium |
| Liu et al.2022 [27] | Unclear | Unclear | High risk | Unclear | Low risk | Low risk | Unclear | Medium |
| Li et al.2021 [29] | Low risk | Unclear | High risk | Unclear | Low risk | Low risk | Low risk | Medium |
| Xu et al.2021 [30] | Low risk | Unclear | High risk | Unclear | Low risk | Low risk | Unclear | Medium |
| Sha et al.2021 31] | Low risk | Unclear | High risk | Unclear | Low risk | Low risk | Unclear | Medium |
| Li et al.2021 [32] | Low risk | Unclear | High risk | Unclear | Low risk | Low risk | High risk | High |
| Bi et al.2020 [33] | Low risk | Unclear | High risk | Unclear | Low risk | Low risk | Unclear | Medium |
| Pang et al.2018 [34] | Low risk | Unclear | High risk | Unclear | Low risk | Low risk | Low risk | Medium |
| Cao et al.2018 [35] | Low risk | Unclear | High risk | Unclear | Low risk | Low risk | Unclear | Medium |
| Sun et al.2017 [36] | Low risk | Unclear | High risk | Unclear | Low risk | Low risk | Unclear | Medium |
| Li et al.2017 [37] | Low risk | Unclear | High risk | Unclear | Low risk | Low risk | Low risk | Medium |
| Luo et al.2017 [38] | Unclear | Unclear | High risk | Unclear | Low risk | Low risk | Unclear | Medium |
| Kuang et al.2017 [39] | Unclear | Unclear | High risk | Unclear | Low risk | Low risk | Unclear | Medium |
| Zhang et al.2017 [40] | Low risk | Unclear | High risk | Unclear | Low risk | Low risk | Unclear | Medium |
| Xu et al.2017 [41] | Low risk | Unclear | High risk | Unclear | Low risk | Low risk | Unclear | Medium |
| Zhu et al.2017 [42] | Low risk | Unclear | High risk | Unclear | Low risk | Low risk | Unclear | Medium |
| Zhao et al.2015 [44] | Unclear | Unclear | High risk | Unclear | Low risk | Low risk | High risk | High |
| Li et al.2014 [45] | Unclear | Unclear | High risk | Unclear | Low risk | Low risk | Unclear | Medium |
| Chen et al.2013 [50] | Low risk | Unclear | High risk | Unclear | Low risk | Low risk | Low risk | Medium |
| Xiang et al.2008 [53] | Low risk | Unclear | High risk | Unclear | Low risk | Low risk | Unclear | Medium |
| Lin et al.2007 [54] | High risk | High risk | High risk | High risk | Low risk | Low risk | Unclear | High |
| Chen et al.2009 [55] | High risk | High risk | High risk | High risk | Low risk | Low risk | Unclear | High |
| Zhang et al.2009 [56] | High risk | High risk | High risk | High risk | Low risk | Low risk | Low risk | High |

**Quality assessment of cohort study (Newcastle—Ottawa quality assessment scale)**

| Author | Selection | | | | Comparability | Outcome | | | |
|---|---|---|---|---|---|---|---|---|---|
| | Representativeness of the exposed cohort (1 point) | Selection of the non exposed cohort (1 point) | Ascertainment of exposure (1 point) | Demonstration that outcome of interest was not present at start of study (1 point) | Comparability of cohorts on the basis of the design or analysis (2 point) | Assessment of outcome (1 point) | Was follow-up long enough for outcomes to occur (1 point) | Adequacy of follow up of cohorts (1 point) | Quality (9 point) |

*(Continued)*

**Table 2.** (Continued)

| | | | | | | | | | |
|---|---|---|---|---|---|---|---|---|---|
| Shang et al. 2021 [28] | 1 | 1 | 1 | 1 | 2 | 0 | 1 | 0 | 7 |
| Zhang et al. 2017 [43] | 1 | 1 | 1 | 0 | 2 | 0 | 1 | 1 | 7 |
| Ou et al. 2014 [46] | 1 | 1 | 1 | 0 | 2 | 0 | 1 | 1 | 7 |
| Ning et al. 2014 [47] | 1 | 1 | 1 | 0 | 2 | 0 | 1 | 1 | 7 |
| Zhang et al. 2014 [48] | 1 | 1 | 1 | 0 | 2 | 0 | 1 | 0 | 6 |
| Tang et al. 2013 [49] | 1 | 1 | 1 | 0 | 2 | 0 | 1 | 1 | 7 |
| Ju et al. 2013 [51] | 1 | 1 | 1 | 0 | 2 | 0 | 1 | 1 | 7 |
| Liang et al. 2012 [52] | 1 | 1 | 1 | 0 | 2 | 0 | 1 | 1 | 7 |
| Tang et al. 2009 [57] | 1 | 1 | 1 | 0 | 2 | 0 | 1 | 1 | 7 |
| Liu et al. 2008 [58] | 1 | 1 | 1 | 0 | 2 | 0 | 1 | 1 | 7 |
| Li et al. 2007 [59] | 1 | 1 | 1 | 0 | 2 | 0 | 1 | 0 | 6 |
| Du et al. 2003 [60] | 1 | 1 | 1 | 0 | 2 | 0 | 1 | 1 | 7 |
| Shen et al. 2003 [61] | 1 | 1 | 1 | 0 | 2 | 0 | 1 | 1 | 7 |
| Yu et al. 2002 [62] | 1 | 1 | 1 | 0 | 2 | 0 | 1 | 1 | 7 |
| Zhang et al. 2001 [63] | 1 | 1 | 1 | 0 | 2 | 0 | 1 | 0 | 6 |

## Sensitivity analysis and subgroup analysis

Sensitivity analysis showed the stability of all the outcomes including the recurrence rate of CSC, BCVA, CRT, and adverse event rate.

For subgroup analyses, the heterogeneity of BCVA versus CRT outcomes decreased when limiting the duration of the intervention ($\geq$2 Months), suggesting that the duration of the intervention was a source of heterogeneity. In addition, the heterogeneity of BAVC outcomes was significantly reduced when the type of restriction was an observational study. Subgroup analyses according to the type of CSC showed altered outcomes involving patients with mixed types of CSC, suggesting that the difference in recurrence rates between the TCM and conventional treatment groups was not statistically significant (OR = 0.51, 95% CI: 0.25,1.07; $I^2$ = 65%). See Tables 3 and 4 for details.

**Narrative description.** One study described SRF [38] and showed that SRF was 28.27 ± 18.52 d in the herbal group and 106.04 ± 83.38 d in the control group, which was statistically significant (p < 0.01).

**Publication bias evaluation.** The funnel plot based on recurrence rates is shown in Fig 6, with the majority of studies located in the upper middle of the funnel. Egger's regression test was used to detect publication bias, and the result showed P = 0.052, indicating no significant publication bias (S1 Fig).

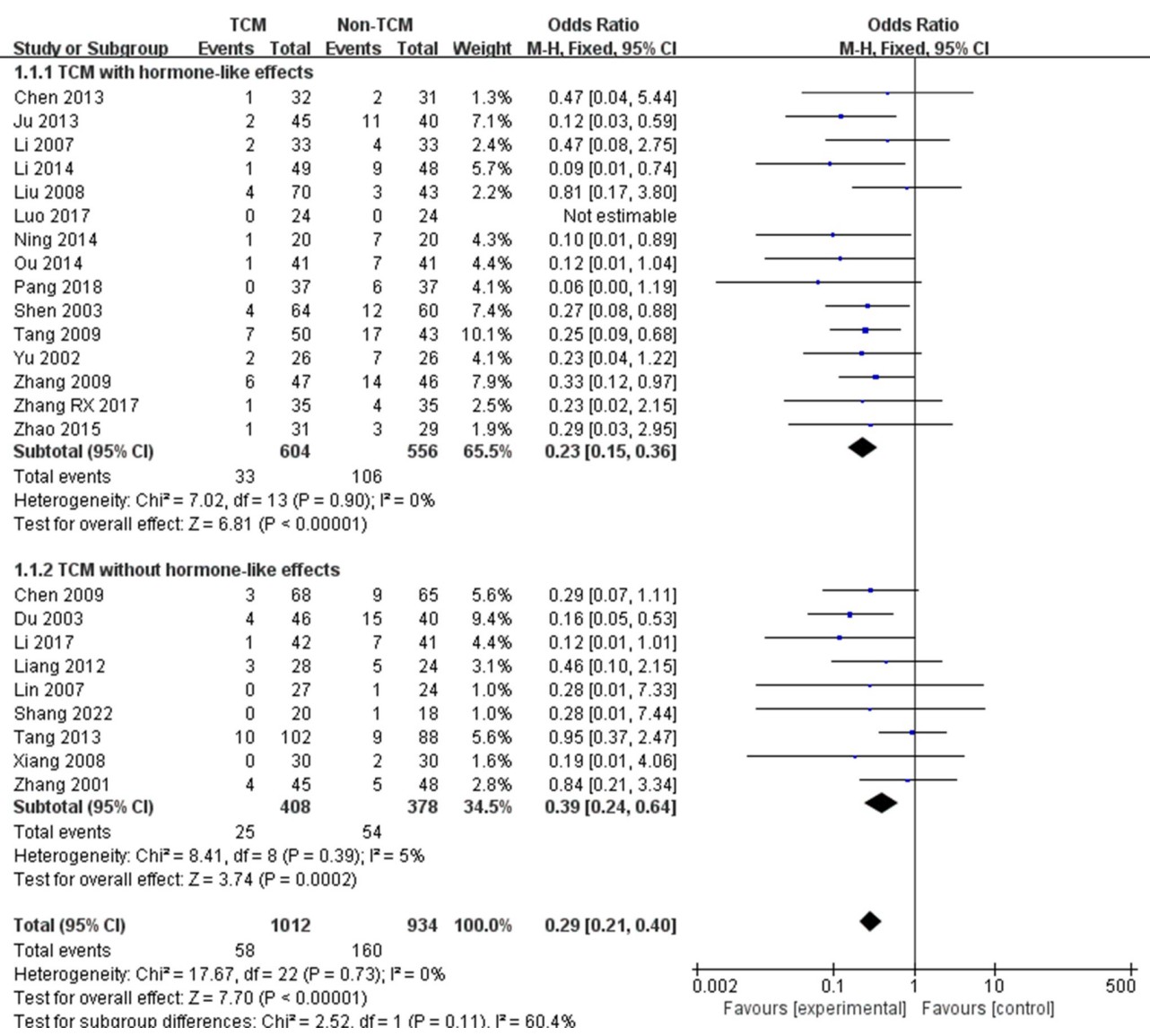

**Fig 2. The meta-analysis results of recurrence rate.**

## Discussion

The present meta-analysis showed that TCM did not trigger the risk of CSC recurrence; on the contrary, compared with conventional treatment, TCM treatment could reduce the recurrence of CSC. In addition, the results of the meta-analysis suggested that TCM had the effect of reducing CRT and improving BCVA, and had no significant side effects compared with conventional treatment. However, limited by the quality of included studies, the certainty of outcomes is a concern.

The use of corticosteroids is the most significant external risk factor for developing CSC, with odds ratios as high as 37:1 being reported [64]. Although rare, in some cases even minimal exposure to corticosteroids exposure has been associated with an increased risk, exacerbation, and/or recurrence of CSC [65, 66], suggesting that the increased risk of developing CSC is not solely dependent on the dose or mode of corticosteroid administration, but may also

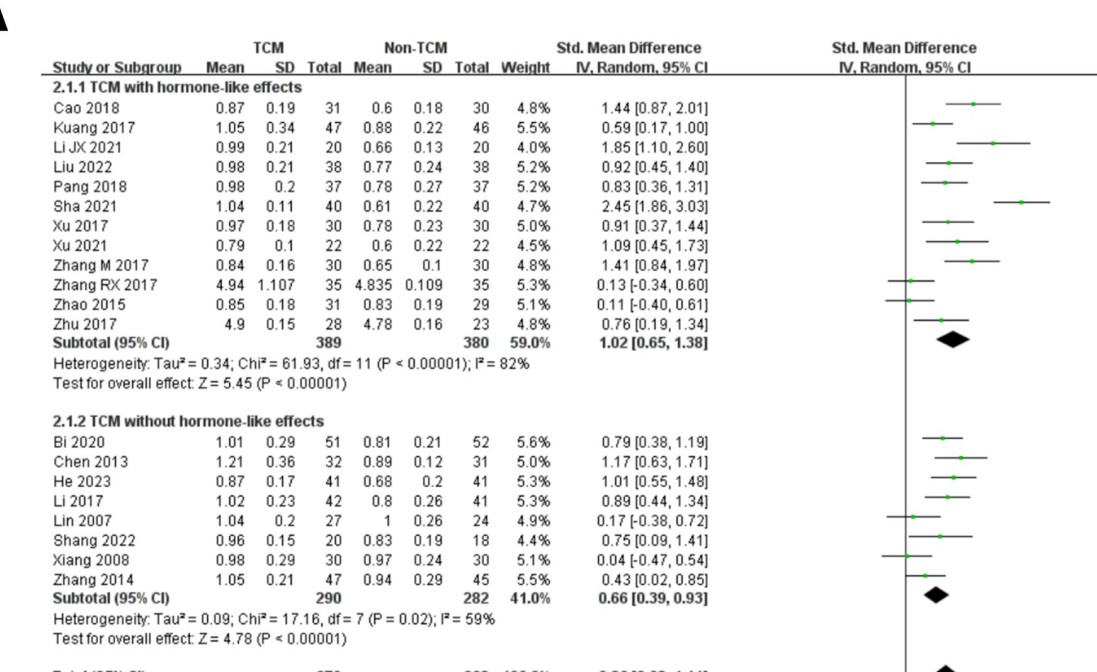

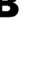

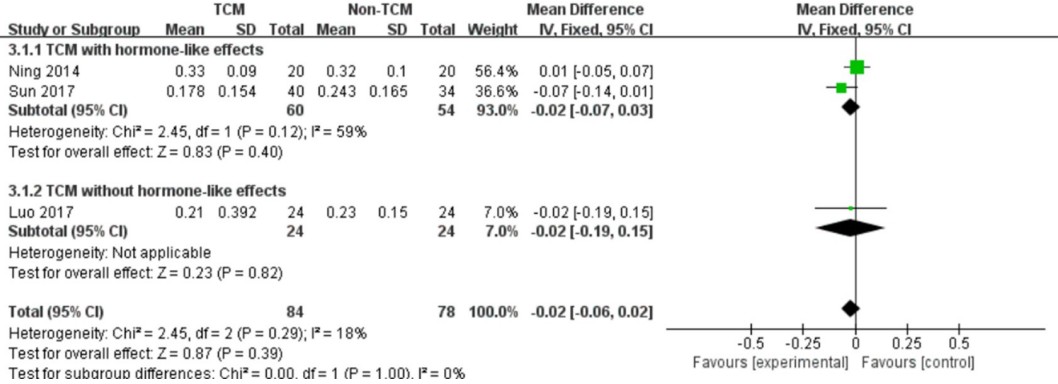

**Fig 3. The meta-analysis results of BCVA.** (A) BCVA (standard visual acuity chart / 5SL chart); (B) BCVA (logMAR).

depend on genetic predisposition and/or an increased vulnerability to corticosteroid exposure in some individuals [9]. The mechanism of corticosteroid-induced CSC may be related to the activation of both the gluco- and the mineralocorticoid (MR) receptors. As MR over-activation is pathogenic in the retina and choroid, it could mediate the pathogenic effects of corticosteroids in CSC [67].

Some herbal medicines have been feared to cause recurrence and exacerbation of CSC because of their hormonal content. This study did not find evidence that herbal medicines induced CSC recurrence, even those containing hormonal components. There are several possible reasons for this: hormone-containing herbs account for a relatively small percentage of the components in the formula, and the content or activity of the hormone components is

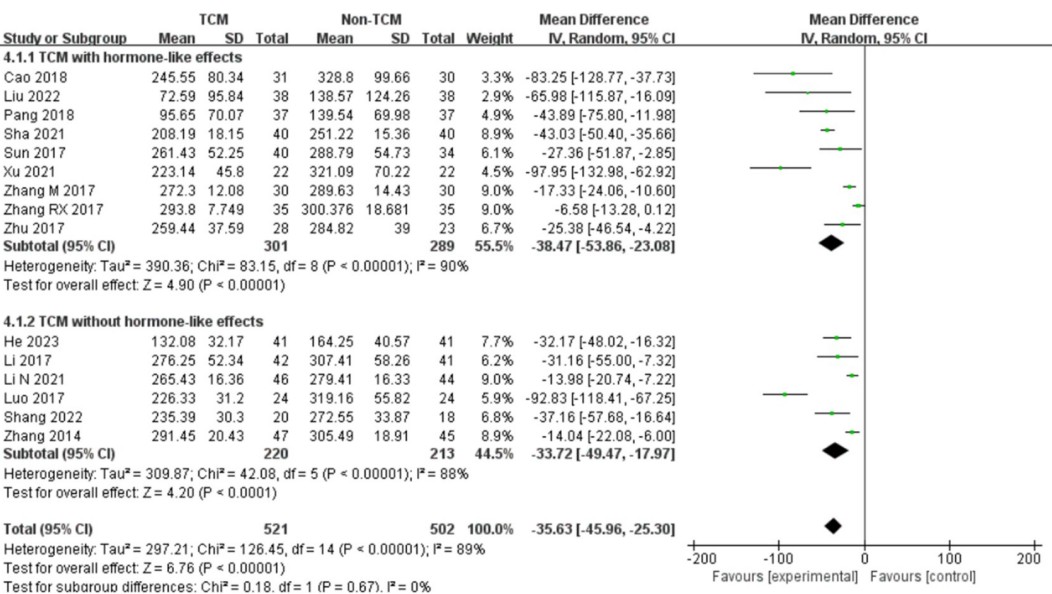

**Fig 4. The meta-analysis results of CRT.**

disturbed during heating and boiling or further processing [68]. In addition, some herbs mostly play a hormone-like pharmacological role, such as licorice [69] and ginseng [70]. Another possible reason is that Chinese medicines contain such a small amount of hormonal components that they cannot trigger significant side effects. Besides, the pharmacological actions of herbs are complex, and the interactions of multiple targets of action among individual drugs in the herbal formulas further interfere with the hormonal effects [71, 72].

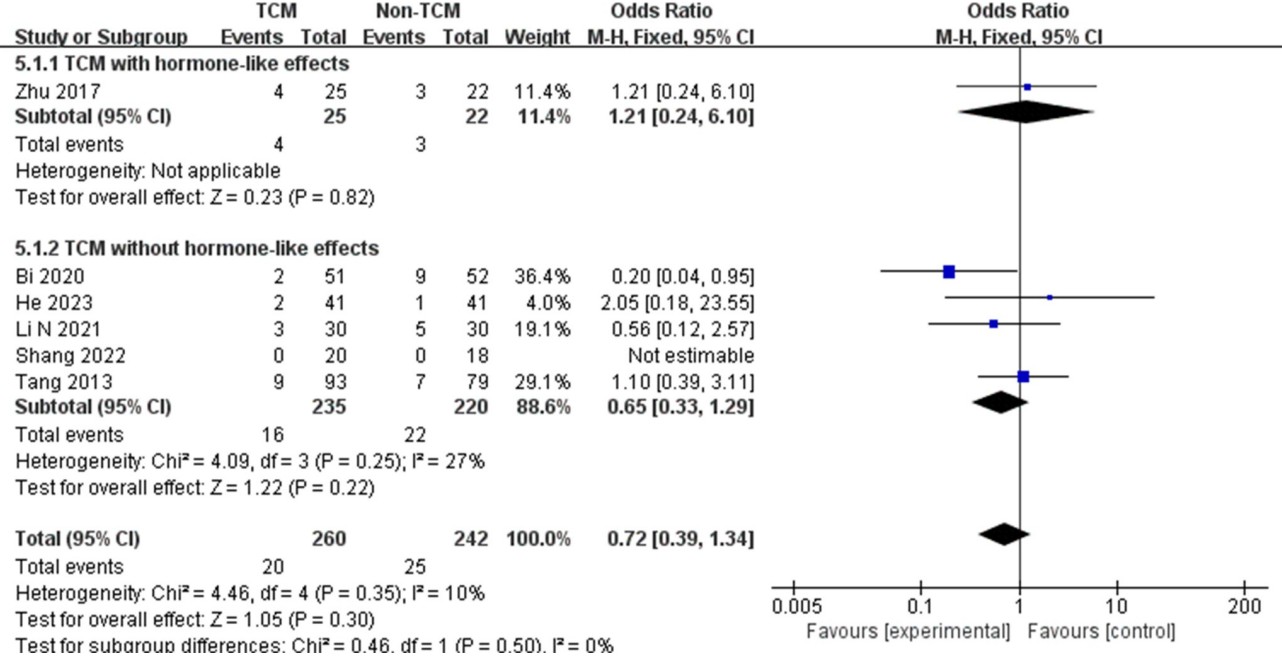

**Fig 5. The meta-analysis results of adverse events.**

**Table 3. Subgroup analysis.**

| | Subgroup analysis | No. of trials (reference nos.) | No. of eyes | $I^2$ | OR [95% CI] | P value |
|---|---|---|---|---|---|---|
| **Recurrence rate** | Overall | 24 (28,34,37,38,43–47,49–63) | 1946 | 0 | 0.29 [0.21, 0.40] | <0.0001 |
| | Study design | | | | | |
| | RCT | 10 (34,37,38,44,45,50,53–56) | 762 | 0 | 0.22 [0.12, 0.40] | <0.0001 |
| | Observational study | 14 (28,43,46,47,49,51,52,57–63) | 1184 | 6% | 0.33 [0.22, 0.47] | <0.0001 |
| | CSC type | | | | | |
| | Acute CSC | 14 (37,38,43,45,47,49,50,53–56,59,62,63) | 1139 | 0 | 0.35 [0.23, 0.54] | <0.0001 |
| | Chronic CSC | 1 (34) | 74 | - | 0.06 [0.00, 1.09] | 0.07 |
| | Mixed (acute+chronic) CSC | 4 (52,57,58,60) | 344 | 2% | 0.29 [0.16, 0.54] | <0.0001 |
| | Intervention course | | | | | |
| | < 2 Months | 12 (28,38,43,47,49,50,53–57,59) | 945 | 0 | 0.36 [0.23, 0.57] | <0.0001 |
| | ≥2 Months | 9 (34,37,44–46,51,52,60,61) | 743 | 0 | 0.17 [0.1, 0.3] | <0.0001 |
| | Intervention type 1 | | | | | |
| | vitamin supplements/ retinal microcirculation-improving drugs | 18 (34,37,44–47,49–52,55,56,58–63) | 1571 | 1% | 0.29 [0.21, 0.41] | <0.0001 |
| | Laser therapy | 5 (28,43,53,54,57) | 327 | 0 | 0.27 [0.13, 0.58] | 0.0008 |
| | No treatmeent | 1 (38) | 48 | - | - | - |
| | Intervention type 2 | | | | | |
| | TCM alone | 5 (38,45,47,52,55) | 370 | 0 | 0.21 [0.09, 0.48] | 0.0002 |
| | TCM+C | 19 (28,34,37,43,44,46,49–51,53,54,56–63) | 1576 | 0 | 0.31 [0.22, 0.43] | <0.0001 |

| | Subgroup analysis | No. of trials (reference nos.) | No. of eyes | $I^2$ | SMD [95% CI] | P value |
|---|---|---|---|---|---|---|
| **BCVA (decimal VA/ 5SL)** | Overall | 20 (26–28,30–35,37,39,40–44,48,50,53,54) | 1341 | 77% | 0.86 [0.62, 1.11] | <0.0001 |
| | Study design | | | | | |
| | RCT | 17 (26,27,30–35,37,39,40,42,44,50,53,54) | 1141 | 77% | 0.94 [0.68, 0.21] | <0.0001 |
| | Observational study | 3 (28,43,48) | 200 | 14% | 0.39 [0.08, 0.69] | 0.01 |
| | CSC type | | | | | |
| | Acute CSC | 12 (26,27,30,33,37,40–43,50,53,54) | 803 | 63% | 0.77 [0.53, 1.01] | <0.0001 |
| | Chronic CSC | 6 (31,32,34,35,39,48) | 440 | 88% | 1.23 [0.63, 1.84] | <0.0001 |
| | Intervention course | | | | | |
| | < 2 Months | 11 (28,31,33,35,39,40,43,48,50,53,54) | 771 | 85% | 0.84 [0.14, 1.23] | <0.0001 |
| | ≥2 Months | 8 (26,27,30,34,37,41,42,44) | 530 | 25% | 0.81 [0.6, 1.02] | <0.0001 |
| | Intervention type 1 | | | | | |
| | vitamin supplements/ retinal microcirculation-improving drugs | 15 (26,27,30–35,37,39,40–42,44,50) | 1030 | 73% | 1.08 [0.80, 1.31] | <0.0001 |
| | Laser therapy | 5 (28,43,48,53,54) | 311 | 0 | 0.28 [0.05, 0.50] | 0.01 |
| | Intervention type 2 | | | | | |
| | TCM alone | 4 (30,32,35,41) | 205 | 36% | 1.28 [0.89, 1.66] | <0.0001 |
| | TCM+C | 16 (26–28,31,33,34,37,39,40,42–44,48,50,53,54) | 1136 | 78% | 0.77 [0.50, 1.03] | <0.0001 |

| | Subgroup analysis | No. of trials (reference nos.) | No. of eyes | $I^2$ | MD [95% CI] | P value |
|---|---|---|---|---|---|---|

(*Continued*)

Table 3. (Continued)

| CRT | Overall | 15 (26–31,34–38,40,42,43,48) | 1023 | 89% | -35.63 [-45.96,-25.30] | <0.0001 |
| --- | --- | --- | --- | --- | --- | --- |
| | Study design | | | | | |
| | RCT | 12 (26,27,29–31,34–38,40,42) | 823 | 88% | -42.14 [-54.89,-29.40] | <0.0001 |
| | Observational study | 3 (28,43,48) | 200 | 76% | -15.51 [-27.41,-3.61] | 0.01 |
| | CSC type | | | | | |
| | Acute CSC | 10 (26,27,29,30,36–38,40,42,43) | 678 | 88% | -34.19 [-46.59,-21.79] | <0.0001 |
| | Chronic CSC | 4 (31,34,35,48) | 307 | 91% | -39.7[-62.79,-16.6] | 0.0008 |
| | Intervention course | | | | | |
| | < 2 Months | 9 (28,29,31,35,36,38,40,43,48) | 613 | 92% | -31.09 [-43.34,-18.85] | <0.0001 |
| | ≥2 Months | 6 (26,27,30,34,37,42) | 410 | 66% | -44.65 [-63.31,-25.98] | <0.0001 |
| | Intervention type 1 | | | | | |
| | vitamin supplements/retinal microcirculation-improving drugs | 10 (26,27,29,31,34–37,40,42) | 731 | 82% | -31.74 [-42.57,-20.92] | <0.0001 |
| | Laser therapy | 4 (28,30,43,48) | 244 | 90% | -30.58 [-50.44,-10.72] | 0.003 |
| | No treatment | 1 (38) | 48 | - | -92.83 [-118.41,-67.25] | <0.0001 |
| | Intervention type 2 | | | | | |
| | TCM alone | 4 (29,35,36,38) | 273 | 93% | -51.93 [-92.68,-11.18] | 0.01 |
| | TCM+C | 11 (26–28,30,31,34,37,40,42,43,48) | 750 | 88% | -31.49 [-43.29,-20.59] | <0.0001 |
| Adverse event | Subgroup analysis | No. of trials (reference nos.) | No. of patients | I² | OR [95% CI] | P value |
| | Overall | 6 (26,28,29,33,42,49) | 502 | 10% | 0.72 [0.39, 1.34] | 0.30 |
| | Study design | | | | | |
| | RCT | 4 (26,29,33,42) | 292 | 18% | 0.56 [0.25, 1.24] | 0.14 |
| | Observational study | 2 (28,49) | 210 | - | 1.10 [0.39, 3.11] | 0.85 |
| | CSC type | | | | | |
| | Acute CSC | 4 (26,29,33,49) | 417 | 27% | 0.65 [0.33, 1.29] | 0.22 |
| | Chronic CSC | 1 (42) | 47 | - | 1.21 [0.24, 6.10] | 0.82 |
| | Intervention course | | | | | |
| | < 2 Months | 4 (28,29,33,49) | 373 | 39% | 0.59 [0.29, 1.20] | 0.15 |
| | ≥2 Months | 2 (26,42) | 129 | 0 | 1.43 [0.37, 5.46] | 0.60 |
| | Intervention type 1 | | | | | |
| | vitamin supplements/ retinal microcirculation-improving drugs | 5 (26,29,33,42,49) | 464 | 0% | 0.72 [0.39, 1.34] | 0.30 |
| | Laser therapy | 1 (28) | 38 | - | - | - |
| | Intervention type 2 | | | | | |
| | TCM alone | 1 (29) | 60 | - | 0.56 [0.12, 2.57] | 0.45 |
| | TCM+C | 5 (26,28,33,42,49) | 442 | 30% | 0.76 [0.38, 1.49] | 0.42 |

The outcome of the meta-analysis of CRT shows that Chinese herbs could significantly reduce the thickness of retinal edema in patients with CSC, although the outcome is highly heterogeneous. Similarly, several studies have identified the absorption-promoting effects of

**Table 4. Sensitivity analyses.**

| | Sensitivity analyses | No. of included trials (reference nos.) | OR [95% CI] | P |
|---|---|---|---|---|
| **Recurrence rate** | Overall analysis | 24 (28,34,37,38,43–47,49–63) | 0.30 [0.22, 0.41] | <0.0001 |
| | Excluding studies with low quality | 18 (28,34,37,38,44–47,49–52,56,57,59–61) | 0.29 [0.2, 0.41] | <0.0001 |
| | Excluding small trial (participants < 50) | 21 (34,37,44–46,49–63) | 0.31 [0.22, 0.43] | <0.0001 |
| | Excluding the lagest trial | 23 (28,34,37,38,44–47,49–63) | 0.26 [0.18, 0.36] | <0.0001 |
| | Excluding studies containing Chinese patent medicine | 22 (28,34,38,44–47,49–63) | 0.26 [0.19, 0.37] | <0.0001 |
| | Using random-effects model | 24 (28,34,37,38,43–47,49–63) | 0.32 [0.23, 0.45] | <0.0001 |

| | Sensitivity analyses | No. of included trials (reference nos.) | SMD [95% CI] | P |
|---|---|---|---|---|
| **BCVA (decimal VA/5SL)** | Overall analysis | 20 (26–28,30–35,37,39,40–44,48,50,53,54) | 0.86 [0.62, 1.11] | <0.0001 |
| | Excluding studies with low quality | 18 (26–28,30–35,37,39,40–44,48,50) | 0.95 [0.70, 1.19] | <0.0001 |
| | Excluding small trial (participants < 50) | 17 (26,27,31,33–35,37,39,40–44,48,50,53,54) | 0.81 [0.55 1.07] | <0.0001 |
| | Excluding the lagest trial | 19 (26–28,30–32,34,35,37,39,40–44,48,50,53,54) | 0.87 [0.61, 1.13] | <0.0001 |
| | Excluding studies containing Chinese patent medicine | 19 (26–28,30–35,39,40–44,48,50,53,54) | 0.86 [0.61, 1.12] | <0.0001 |
| | Using fixed-effects model | 20 (26–28,30–35,37,39,40–44,48,50,53,54) | 0.81 [0.69, 0.92] | <0.0001 |

| | Sensitivity analyses | No. of included trials (reference nos.) | MD [95% CI] | P |
|---|---|---|---|---|
| **CRT** | Overall analysis | 15 (26–31,34–38,40,42,43,48) | -35.63 [-45.96,-25.30] | <0.0001 |
| | Excluding studies with low quality | 13 (26–31,34–38,40,42) | -41.57 [-53.55,-29.60] | <0.0001 |
| | Excluding small trial (participants < 50) | 13 (26,27,29,31,34–38,40,42,43,48) | -31.88 [-42.25,-21.51] | <0.0001 |
| | Excluding the lagest trial | 14 (26–31,34–38,40,42,43) | -38.26 [-49.70,-26.81] | <0.0001 |
| | Excluding studies containing Chinese patent medicine | 14 (26–31,34–36,38,40,42,43,48) | -36.02 [-46.80,-25.25] | <0.0001 |
| | Using fixed-effects model | 15 (26–31,34–38,40,42,43,48) | -21.91[-24.84,-18.98] | <0.0001 |

| | Sensitivity analyses | No. of included trials (reference nos.) | OR [95% CI] | P |
|---|---|---|---|---|
| **Adverse event** | Overall analysis | 6 (26,28,29,33,42,49) | 0.72 [0.39, 1.34] | 0.30 |
| | Excluding studies with low quality | 5 (26,28,29,33,42) | 0.56 [0.25, 1.22] | 0.15 |
| | Excluding small trial (participants < 50) | 5 (26,29,33,42,49) | 0.72 [0.39, 1.34] | 0.30 |
| | Excluding the lagest trial | 5 (26,28,29,33,42) | 0.56 [0.25, 1.22] | 0.15 |
| | Excluding studies containing Chinese patent medicine | 5 (26,28,29,33,42) | 0.56 [0.25, 1.22] | 0.15 |
| | Using random-effects model | 6 (26,28,29,33,42,49) | 0.76 [0.37, 1.53] | 0.44 |

TCM on fundus edema [14, 73, 74], and this process may involve multiple mechanisms. Choroidal dysfunction is considered to be the main etiology of CSC, and venous congestion, inflammation, and hemodynamic changes can lead to choroidal hyperpermeability and subsequent fluid leakage in CSC [75]. TCM has been found to improve fundus microcirculation and inhibit inflammation [74, 76–79], which may reduce fluid leakage and facilitate the absorption of edema. In addition, the activation of mineralocorticoid receptors can lead to recurrence and exacerbation of CSC [67], and there have been studies showing the modulation of mineralocorticoid activity by a variety of herbal medicines, which may also be one of the mechanisms of action of TCM in the treatment of CSC [80–82].

Our BCVA and CRT outcomes were highly heterogeneous, and subgroup analyses showed that part of the heterogeneity came from the type of study and the duration of the treatment intervention. We noted that the included studies used 3 visual acuity counting methods, including decimal VA, 5SL, and logMAR. Among them, decimal VA and 5SL charts account for the majority of the included studies (decimal VA,18/23; 5SL, 2/23). In China, both decimal VA and 5SL charts are commonly used in screening, ophthalmology clinic. However, there are some differences between the two approaches and the logMAR recording method. Decimal charts have an irregular progression in size and are often truncated, especially in the lines testing low acuity, whereby only one or three optotypes are presented per line. Thus, the data may

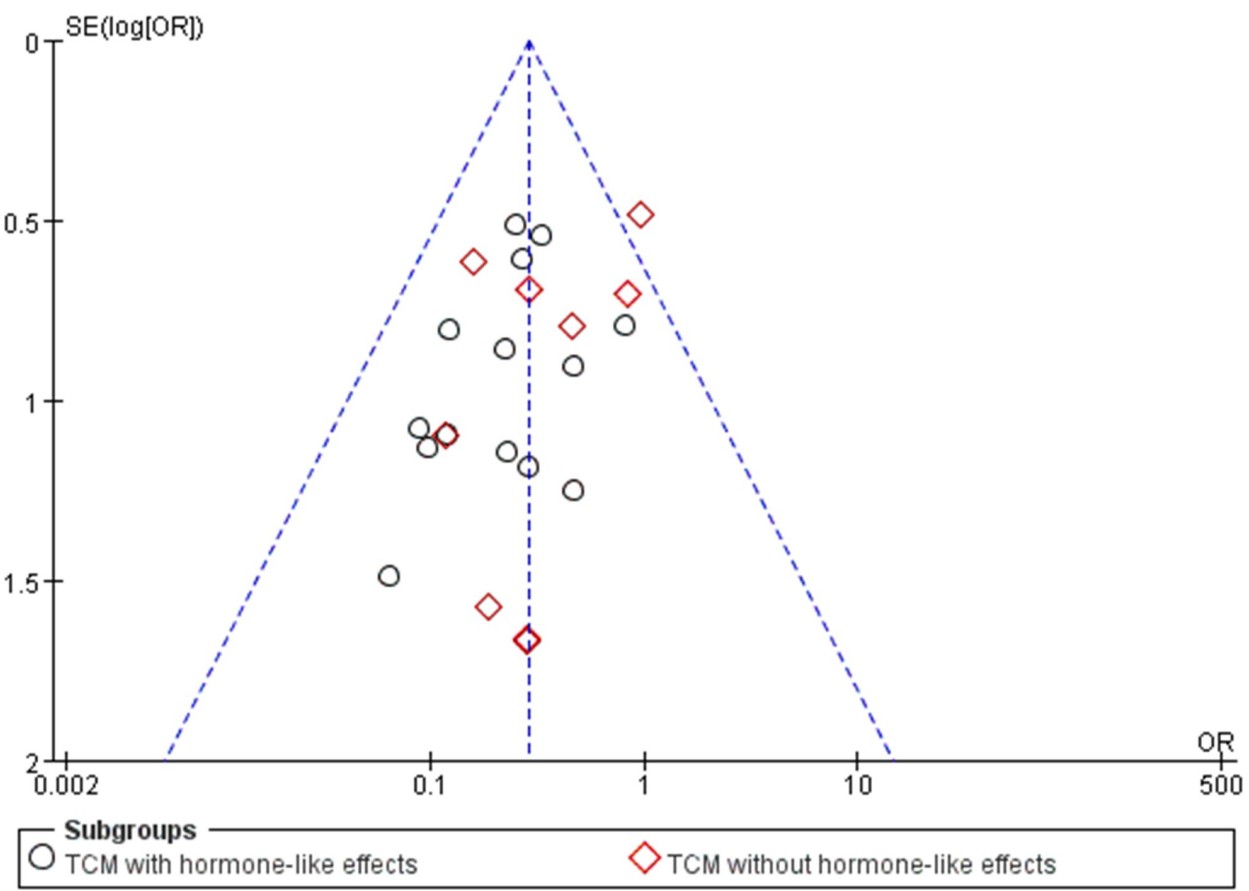

**Fig 6. The funnel plot of included studies.**

not follow a normal distribution, and this problem is not overcome merely by converting the data to logMAR [83]. The design of the 5SL chart follows the Weber-Fechner rule, which was thought could be directly used for VA statistics and efficacy evaluation, and be essentially equivalent to the logMAR recording method [84]. However, the agreement between 5SL and logMAR is not high, and the VA measured by the 5SL chart is slightly better than that by the logMAR vision chart [84, 85]. Thus, the different methods of recording visual outcomes may contribute to the heterogeneity of BCVA. In addition, the heterogeneity of BCVA decreased significantly when limiting the intervention to TCM alone, suggesting that the different types of intervention may also be a source of heterogeneity in BCVA outcomes.

In terms of recurrence rate, only one study included patients with chronic CSC, and the results showed no significant difference between the TCM and conventional treatment groups. However, the small sample size of the study greatly limited the certainty of this outcome. Similarly, when limiting the use of the logMAR method to document BCVA, it was found that herbal medicines did not suggest an improvement in BCVA, and this outcome remains limited by the insufficient number of studies and patients.

Our study had a comprehensive search strategy that included all the literature on herbal medicine-related treatments for CSC to the best of our knowledge. The sensitivity analysis suggested that the outcome of the meta-analysis was stable. Nonetheless, we have the following limitations: first, although we implemented an adequate and detailed search strategy, the possibility of publication bias cannot be ruled out, which means that some result values may be

**Table 5. Summary of findings.**

| Outcomes | Risk of bias | Inconsistency | Indirectness | Imprecision | Other considerations | Certainty of the evidence |
|---|---|---|---|---|---|---|
| Recurrence rate | Serious[a] | Not serious | Not serious | Not serious | None | ⊕⊕⊕◯ MODERATE |
| BCVA (decimal VA/5SL)) | Serious[a] | Very serious[b] | Not serious | Not serious | None | ⊕◯◯◯ VERY LOW |
| BCVA (LogMAR) | Serious[a] | Not serious | Not serious | Very serious[c] | None | ⊕◯◯◯ VERY LOW |
| CRT | Serious[a] | Very serious[d] | Not serious | Not serious | None | ⊕⊕◯◯ LOW |
| Number of adverse events | Serious[a] | Not serious | Not serious | Very serious[c] | None | ⊕◯◯◯ VERY LOW |

**GRADE Working Group grades of evidence**

**High certainty:** We are very confident that the true effect lies close to that of the estimate of the effect.

**Moderate certainty:** We are moderately confident in the effect estimate: The true effect is likely to be close to the estimate of the effect, but there is a possibility that it is substantially different.

**Low certainty:** Our confidence in the effect estimate is limited: The true effect may be substantially different from the estimate of the effect.

**Very low certainty:** We have very little confidence in the effect estimate: The true effect is likely to be substantially different from the estimate of effect.

Explanations

Concerns about bias in the domains of allocation concealment, blinding of outcome assessment and selective reporting.

b. I-squared = 77%.

c. Very concerned about the number of studies and participants.

d. I-squared = 89%.

amplified, especially in the presence of selective reporting bias in some included studies. Second, the inclusion population of this study was all Chinese, which is restrictive for generalization to other populations. In addition, the therapeutic measures in the control group included in the study were mainly improvement of microcirculation, vitamin supplements, laser, and other therapeutic measures, of which there were fewer studies containing laser therapy and no studies involving PDT and anti-VEGF therapy, the effect of Chinese medicine in this population remains uncertain. Last, studies involving BCVA and CRT were at high risk of combined intervention bias (performance bias), inconsistency (high heterogeneity), and imprecision (small samples), limiting the quality of evidence. As a result, no evidence was highly definitive. According to the GRADE evaluation system, the quality of our evidence ranged from "moderate" to "very low" (Table 5).

Overall, our findings suggest that herbal medicines do not increase the risk of CSC recurrence; rather, the combination of herbal medicines may play a role in decreasing the rate of CSC recurrence and improving BCVA and CRT in patients with CSC compared with conventional treatment.

## Supporting information

**S1 Fig. Egger's regression test.** Egger's regression test based on recurrence rates.
(DOC)

**S1 File. PRISMA 2020 checklist.**
(DOCX)

**S2 File. Raw data for meta-analysis.**
(XLS)

**S1 Table. Search strategy.**
(DOC)

**S2 Table. Evaluation criteria for the overall risk of bias in randomized controlled trials.**
(DOC)

# Acknowledgments

All authors of this manuscript would like to express our sincere gratitude to the stuffs of China Academy of Chinese Medical Sciences for their advice on the selection of Chinese medicines containing hormone-like effects.

# Author Contributions

**Conceptualization:** Wu Sun, Liqun Chu.

**Data curation:** Shuting Ru, Jian Sun, Hang Shi, Yu Liang, Jianguo Wu, Wu Sun.

**Writing – original draft:** Wu Sun.

**Writing – review & editing:** Wanyu Zhou, Dong Wei.

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
