## [Decision Letter · Decision Letter 0]

8 Feb 2024

PONE-D-23-26616Effects of Traditional Chinese medicine  in the treatment of patients with central serous chorioretinopathy: a systematic review and meta-analysisPLOS ONE

Dear Dr. Sun,

Thank you for submitting your manuscript to PLOS ONE. After careful consideration, we feel that it has merit but does not fully meet PLOS ONE’s publication criteria as it currently stands. Therefore, we invite you to submit a revised version of the manuscript that addresses the points raised during the review process.

We look forward to receiving your revised manuscript.

Kind regards,

Tungki Pratama Umar, M.D.

Academic Editor

PLOS ONE

Journal Requirements:

"This work was supported by “Major Research Project of Acupuncture and Moxibustion Clinical Discipline, Science and Technology Innovation Project, Chinese Academy of Traditional Chinese Medicine” (No. C12021A03513) and “Independent Selection Project of Chinese Academy of Traditional Chinese Medicine” (No. ZZ16-XRZ-024)."

6. We note that this manuscript is a systematic review or meta-analysis; our author guidelines therefore require that you use PRISMA guidance to help improve reporting quality of this type of study. Please upload copies of the completed PRISMA checklist as Supporting Information with a file name “PRISMA checklist”.

Additional Editor Comments:

Abstract

1. Overall abstract should be rewritten. Use non-structured abstract.

2. Introduction should have more background regarding study importance or motive

3. Methods: Add more about inclusion criteria, used databases, PRISMA adherence, type of meta-analysis, etc

4. Define abbreviations written in it

Introduction

1. Add more information about treatment and prevention of the disease, and why TCM is used

2. Overall, it is still confusing, why only writing bad side of the intervention? Please add more information, especially it is commonly inferred that SR/MA is intended for searching potential treatment/favorable effect (which is also apparent in the authors’ abstract)

Methods

1. For inclusion criteria, write it as narrative (as the exclusion criteria)

2. How about treatment outside of TCM? Is it considered as exclusion criteria or not? Please elaborate more about it

3. “Case series reports” are misleading term, please correct it to case series or case reports

4. Add no full text as exclusion criteria

5. Databases with abbreviation should be written as long form in first occurrence

6. Data extraction: add what data the authors wanted to gather

7. Add initials of all reviewer, not just write one or two

8. Add citation for RevMan (RevMan 5.4.1 (Cochrane Collaboration)

9. Delete “according to the synthesis without the meta-analysis guideline.”

10. Describe reviewers’ credentials for GRADE (are they ophthalmologists, TCM practitioners, or what?)

11. Rewrite and recite all PRISMA (mandatory). The authors were using an outdated version of it (2009). Should use 2020 version.

Result

1. Add criteria about acute and chronic CSC (suggestion: locate in method, data extraction)

2. ROB was used to evaluate the risk of bias in 23 RCTs [10,11,18,20] � but this is only 4 studies?

3. I am not really sure about the association about “funding” that is mentioned in the result (suggestion: delete)

4. Where is GRADE? Not shown in the result

5. For publication bias, do the analysis (e.g. Egger test) to determine whether there is a true bias. If it stays, give more explanation about the possible source.

Discussion

1. Expand discussion section, mostly by giving more detailed explanation about why TCM is beneficial while CS is usually not. Add more substances that may be responsible for it (not only giving general mechanism).

General

1. Please recheck for grammar and typographic error (such as: backgroud)

2. The information is interesting. However, there are several inconsistencies that may hinder further processing of this manuscript.

Reviewers' comments:

Reviewer's Responses to Questions

**Comments to the Author**

1. Is the manuscript technically sound, and do the data support the conclusions?

Reviewer #1: Partly

Reviewer #2: Yes

Reviewer #3: Partly

Reviewer #4: Yes

2. Has the statistical analysis been performed appropriately and rigorously? 

Reviewer #1: No

Reviewer #2: Yes

Reviewer #3: Yes

Reviewer #4: Yes

3. Have the authors made all data underlying the findings in their manuscript fully available?

Reviewer #1: Yes

Reviewer #2: Yes

Reviewer #3: Yes

Reviewer #4: Yes

4. Is the manuscript presented in an intelligible fashion and written in standard English?

Reviewer #1: Yes

Reviewer #2: Yes

Reviewer #3: Yes

Reviewer #4: Yes

5. Review Comments to the Author

Reviewer #1: 1. The authors should include name of Traditional Chinese medicine (TCM) that used in this study, such as wolfberry, licorice etc

2. The authors should clarify the kind of the TCM

"Single TCM such as: wolfberry, etc or combination of the TCM such as: wolfberry + licorice etc"

3. The study is lack of information about the safety of the TCM which is the one of the objective in this study

4. The authors must provide an explanation regarding treatment with TCM + C and TCM alone to avoid data bias

Reviewer #2: Dear authors,

Stress and cortisol are well-established factors in the pathogenesis of CSC. A meta-analysis on the effects of Chinese traditional medicine with purported hormone-like effects is therefore a relevant and interesting topic. I hope my comments will be helpful to further improve the manuscript.

Page 5: In the data analysis section, there is a typographical error regarding CSC type. Chornic is greater than 6 months and acute should be less than 6 months. Also in the next line, ‘no treatment, et al’ seems to be an error.

Page 6: In literature search, please recheck the number of duplicate studies that were removed (995). A total of 1826 articles and 838 remaining, does not match with it. Also, the number of full text articles excluded (321) needs to be rechecked in figure 1.

Page 7: Please mention the hormone-like and non-hormone-like herbal medicines that were used in the included studies. Also what were the vitamins and retinal microcirculation-improving drugs?

Page 11: The mixed type CSC is mentioned. However, the definition is not included in the methods/data analysis. Does it mean CSC in combination with RPE detachments, neovascularization, etc?

Reviewer #3: Although a good review, the reviewer feels that it does not add anything new in the literature.

Maybe focusing on a selected TCMs and looking in depth at the effect of the selected TCMs might be more useful. As it stands it is too broad and unfocused.

Reviewer #4: This manuscript reported meta-analysis results about the efficacy and safety of treatments with Traditional Chinese medicine added in treating patients with Central serous chorioretinopathy (CSC). PRISMA guideline is followed. Subgroup analysis and many sensitivity analysis were performed for scientific rigor. However, there are some concerns:

1. It is unclear if all analysis had treated two eyes from the same patient as independent. For example, # of events and # of totals in Figure 2 and Figure 5 are on the patient-level or the eye-level? If those are on the eye-level, the calculation of corresponding OR should take into account the dependence of both eyes from the same patient. Similar concern for the estimation of mean differences exists.

2. It is unclear how standard mean difference is defined for BCVA where 18 studies used a standard visual acuity chart (decimal) and 2 used a logarithmic visual acuity chart (5-point scale). More details should be given.

3. It is recommended that the study team gets a professional biostatistician's help in result explanation. For example, the abstract says " it has no obvious side effects (OR=0.82, 95% CI:0.44,1.52; I2 = 30%)." which should be corrected because the quoted OR only suggested that TCM had no significantly different risk of side effect compared with the control group. Also it is claimed that "This study shows that the use of TCM does not cause the recurrence of CSC" which is not a rigorous statement.

4. Section 2.2.4 listed two primary outcomes. So it is recommended to report 97.5%CI for each primary outcome instead of 95% CIs.

6. PLOS authors have the option to publish the peer review history of their article (what does this mean?). If published, this will include your full peer review and any attached files.

Reviewer #1: No

Reviewer #2: No

Reviewer #3: No

Reviewer #4: No

---

## [Author Response · Author response to Decision Letter 0]

2 May 2024

Dear Editors and Reviewers: 

Thank you for your letter and the reviewers’ comments concerning our manuscript 

entitled “Effects of Traditional Chinese Medicine in the treatment of patients with Central serous chorioretinopathy: a systematic review and meta-analysis”. Those comments are all valuable and very helpful for revising and improving our paper, as well as the important guiding significance to our research. We have studied the comments carefully and have made corrections which we hope meet with approval. Revised portions are marked in red on the paper. The main corrections in the paper and the responses to the reviewer’s comments are as follows: 

Responds to the reviewer’s comments: 

1. Response to comment: adjust for manuscript style.

Response: Thank you for your suggestion. We changed the style based on the style templates. 

2. Response to comment: include a title page within the main document.

Response: Thank you for your suggestion. We added the title page to the manuscript.

3. Response to comment: deposit the data in the participating repository.

Response: Thank you very much for your advice. At this time, we have no plans to store the data in a participating repository, since all the data for this article is in the paper. Thanks again for your comments.

4. Response to comment: updated Funding Statement.

Response: Thank you for your suggestion. We have adjusted the Funding Statement and added it to the title page. 

5. Response to comment: adjust Supporting Information files.

Response: Thank you for your suggestion. We have updated the supplementary information according to the format template, please refer to page 22 and Supplementary information 1-4 for details.

6. Response to comment: updated “PRISMA checklist”.: 

Response: Thank you for your suggestion. We have changed the PRISM 2009 to PRISMA 2020 checklist. Please refer to S4 for details.

Additional Editor Comments

Abstract

1. Overall abstract should be rewritten. Use a non-structured abstract.

Response: Thank you for your suggestion. As per your request, we have made changes to the abstract section. Please see the abstract section of the manuscript for details.

2. The introduction should have more background regarding the study's importance or motive.

Response: Thank you for your suggestion. We have supplemented the background of the study in the abstract.

3. Methods: Add more about inclusion criteria, used databases, PRISMA adherence, type of meta-analysis, etc.

Response: Thank you for your suggestion. We have added a number of elements to the abstract regarding inclusion criteria, including the types of studies and populations involved, and the databases searched. Please see the abstract section of the manuscript for details.

4. Define abbreviations written in it.

Response: Thank you for your suggestion. We have made adjustments to the abbreviated content of the abstract accordingly.

Introduction

1. Add more information about the treatment and prevention of the disease, and why TCM is used.

Response: Thank you for your suggestion. We have added more information about the treatment and prevention of CSC. In addition, we added the reasons and background for why TCM is used to treat CSC. Thank you for your comments.

2. Overall, it is still confusing, why only write the bad side of the intervention? Please add more information, especially it is commonly inferred that SR/MA is intended for searching potential treatment/favorable effects (which is also apparent in the authors’ abstract).

Response: Thank you for your suggestion. First, we learned that countries/regions other than China, such as Singapore, are particularly concerned about TCM as a risk factor for CSC. In order to determine whether TCM carries this potential risk, we conducted this meta-analysis. In fact, we wanted to observe the effect of TCM on CSC to see if it is beneficial or harmful to CSC. 

Considering your concerns, we have added the possible benefits of herbal medicine in the introduction section and described the purpose of the study at the end of the introduction.

Once again, thank you very much for your comments and suggestions.

Methods

1. For inclusion criteria, write it as narrative (as the exclusion criteria)

Response: We have re-written this part according to your suggestion.

2. How about treatment outside of TCM? Is it considered as exclusion criteria or not? Please elaborate more about it

Response: Thank you for your reminding. In our study, studies in the intervention group that combine TCM with treatment in the control group were included, while studies that combine treatment outside of TCM were excluded. We have added this in the Inclusion and Exclusion Criteria section. Thank you for your comments.

3. “Case series reports” are misleading term, please correct it to case series or case reports

Response: We are very sorry for our incorrect writing, and we have re-written this part according to your suggestion.

4. Add no full text as exclusion criteria

Response: We have added it as exclusion criteria according to your suggestion.

5. Databases with abbreviation should be written as long form in first occurrence

Response: Thank you for your reminding. We have re-written this part according to your suggestion.

6. Data extraction: add what data the authors wanted to gather

Response: We have re-written this part according to the Reviewer’s suggestion.

7. Add initials of all reviewer, not just write one or two

Response: Thank you for your reminding. We have added initials of relevant reviewers according to your suggestion.

8. Add citation for RevMan (RevMan 5.4.1 (Cochrane Collaboration)

Response: Thank you for your reminding. We have added citation for RevMan in the beginning of “data analysis” part.

9. Delete “according to the synthesis without the meta-analysis guideline.”

Response: We have deleted this part according to your suggestion.

10. Describe reviewers’ credentials for GRADE (are they ophthalmologists, TCM practitioners, or what?)

Response: Thank you for your reminding. We have described the reviewers’ credentials for GRADE according to your suggestion.

11. Rewrite and recite all PRISMA (mandatory). The authors were using an outdated version of it (2009). Should use 2020 version.

Response: Thank you for your reminding. We have updated PRISM 2009 to PRISM 2000, please refer to S4 for details.

Once again, thank you very much for your comments and suggestions.

Result

1. Add criteria about acute and chronic CSC (suggestion: locate in method, data extraction)

Response: Thank you for your reminding. We have added criteria about acute and chronic CSC in the “data extraction” part. 

2. ROB was used to evaluate the risk of bias in 23 RCTs [10,11,18,20] � but this is only 4 studies?

Response: We are very sorry for our negligence of this part. We have corrected it and thank you for your reminding. 

3. I am not really sure about the association about “funding” that is mentioned in the result (suggestion: delete)

Response: Thank you for your suggestion. We have deleted it according to your suggestion.

4. Where is GRADE? Not shown in the result.

Response: The GRADE result is in the supplementary file. Considering that it is an overall evaluation of the evidence, we have described it in the final discussion section. We have adapted it to the body of the manuscript. Please refer to Table 5 for details. Thank you for your reminding. 

5. For publication bias, do the analysis (e.g. Egger test) to determine whether there is a true bias. If it stays, give more explanation about the possible source.

Response: Thank you for your suggestion. We have added Egger test for the evaluation of publication bias, and the result showed P=0.052, indicating no significant publication bias. Please refer to the publication Bias section for details. The diagram of egger detection is in supplementary material S3.

Discussion

1. Expand discussion section, mostly by giving more detailed explanation about why TCM is beneficial while CS is usually not. Add more substances that may be responsible for it (not only giving general mechanism).

Response: Thank you for your suggestion. We added the relevant studies between glucocorticoids and CSC, and the possible mechanisms of glucocorticoid-induced CSC in the discussion section. In addition, we discuss in detail the possible mechanism of action of TCM in the treatment of CSC. Please see pages 18 and 19 of the Discussion section for specific details.

General

1. Please recheck for grammar and typographic error (such as: backgroud)

Response: We are very sorry for our incorrect writing. We carefully tested the whole text for spelling and grammar. Thank you for your reminder.

2. The information is interesting. However, there are several inconsistencies that may hinder further processing of this manuscript.

Response: Thank you very much for recognizing and commenting on our study. 

We apologize for some of the inconsistencies in our manuscript. We will do our best to improve the manuscript, and We really appreciate for Editors warm work earnestly and hope that the correction will meet with approval. 

Once again, thank you very much for your comments and suggestions.

Reviewer #1: 

1. Response to comment: The authors should include name of Traditional Chinese medicine (TCM) that used in this study, such as wolfberry, licorice etc.

Response: Thank you for your advice. We added the name of Traditional Chinese medicine (TCM) included in the study. For details, please refer to Table 1.

2.Response to comment: The authors should clarify the kind of the TCM.

Response: Thank you for your advice. We have clarified the herbal composition of the TCM treatment included in the study. Bolded herbal compositions indicate the presence of hormones or hormone-like effects. For details, please refer to Table 1.

3. Response to comment: The study is lack of information about the safety of the TCM which is the one of the objective in this study.

Response: Thank you for your comments. We evaluated the safety of TCM using side effects, as described in the meta-analysis section "Adverse events". However, side effects were described in only 6 studies in our results. We contacted the authors but did not receive a positive response. The available evidence suggests that there was no statistically significant difference in the incidence of adverse events between the TCM group and conventional treatment group (OR=0.72, 95% CI: 0.39,1.34; I2 = 10%), regardless of whether the TCM contained hormonal components or not. (Fig 5).

4. Response to comment: The authors must provide an explanation regarding treatment with TCM + C and TCM alone to avoid data bias.

Response: Thank you for your comments! According to your suggestion, we analyzed the results of meta-analysis for TCM alone and TCM combination treatment (TCM+C) in subgroup analysis. The results showed that compared with conventional treatment, both TCM alone (TCM alone) and the combination of TCM (TCM+C) improved the indicators in terms of CSC recurrence rate, BCVA, and CRT. As for adverse events, compared with conventional treatment, neither TCM alone nor TCM+C showed significant differences. For details, please refer to Table 3. Special thanks to you for your good comments！

Once again, thank you very much for your comments and suggestions.

Reviewer #2: 

1.Response to comment: Page 5: In the data analysis section, there is a typographical error regarding CSC type. Chornic is greater than 6 months and acute should be less than 6 months. Also in the next line, ‘no treatment, et al’ seems to be an error.

Response: We are very sorry for our incorrect writing, and we have re-written this part according to your suggestion.

2.Response to comment: Page 6: In literature search, please recheck the number of duplicate studies that were removed (995). A total of 1826 articles and 838 remaining, does not match with it. Also, the number of full text articles excluded (321) needs to be rechecked in figure 1.

Response: We are very sorry for our mistakes., and we have re-written this part. Please refer to Figure 1 for specific changes. 

Thank you for pointing out these details of our mistakes, which will help us improve our manuscript.

3.Response to comment: Page 7: Please mention the hormone-like and non-hormone-like herbal medicines that were used in the included studies. Also what were the vitamins and retinal microcirculation-improving drugs?

Response: Thank you for your advice. We have added the herbal composition of the TCM treatment included in the study. Bolded herbal compositions indicate the presence of hormones or hormone-like effects. In addition, with reference to your comments, we provide a detailed description of the vitamins and retinal microcirculation-improving drugs in control group. For details, please refer to Table 1.

Thank you for your advice. 

4.Response to comment: Page 11: The mixed type CSC is mentioned. However, the definition is not included in the methods/data analysis. Does it mean CSC in combination with RPE detachments, neovascularization, etc?

Response: Thank you for your comment, and we apologize for not describing this clearly. "Mixed type CSC" refers to a mixture of acute and chronic CSC patients, and did not include acute or chronic patients alone. Because we did not know the exact proportion of patients with acute and chronic CSC and were concerned about biasing the results, we performed separate subgroup analyses for this type of study, as shown in Table 3.

We labeled "mixed type CSC" for further clarification. Thank you for your comments.

Once again, thank you very much for your comments and suggestions.

Reviewer #3: 

1. Response to comment: Maybe focusing on a selected TCMs and looking in depth at the effect of the selected TCMs might be more useful. As it stands it is too broad and unfocused.

Response: Thank you for your comments. Our aim in this literature is to explore the effects of herbal medicines on CSC as a whole. Hormones have been shown to be a risk factor for CSC, and some herbs are considered by many ophthalmologists to be a risk factor for recurrence in patients with CSC due to their hormonal components or hormone-like effects. Some ophthalmologists often ask patients with CSC recurrence whether they have taken herbal medicines, without limiting herbal treatment specifically to one herb.

In order to explore the effect of herbal medicines as a whole on CSC, we summarized all the available evidence related to the treatment of CSC with herbal medicines that we are aware of, in order to provide clinically relevant evidence for reference. In addition, we analyzed separately the herbal medicines containing hormones or hormone-like effects to further focus on the issue of herbal medicine use and CSC recurrence.

We hope that this study can show the impact of herbal medicines, especially hormone-like herbal medicines, on CSC with the widest number of studies and sample size, and patient population. Therefore, we chose such an overall search.

Thank you again for your professional opinion.

Reviewer #4: 

1. Response to comment: It is unclear if all analysis had treated two eyes from the same patient as independent. For example, # of events and # of totals in Figure 2 and Figure 5 are on the patient-level or the eye-level? If those are on the eye-level, the calculation of corresponding OR should take into account the dependence of both eyes from the same patient. Similar concern for the estimation of mean differences exists.

Response: Thank you very much for your professional advice! We had previously based our calculations on the eye alone, both for the BCVA outcome in Figure 2 and the adverse event outcome in Figure 5. However, after your reminder, we noticed that the adverse reactions were not about the eyes, but for systemic adverse reactions, including nausea, gastrointestinal reactions, panic, and subcutaneous hemorrhage and hardening accompanied by subcutaneous injection of drugs.

Therefore, we recalculated the adverse events item and conduct

---

## [Decision Letter · Decision Letter 1]

22 May 2024

Effects of Traditional Chinese Medicine  in the treatment of patients with central serous chorioretinopathy: a systematic review and meta-analysis

PONE-D-23-26616R1

Dear Dr. Sun,

We’re pleased to inform you that your manuscript has been judged scientifically suitable for publication and will be formally accepted for publication once it meets all outstanding technical requirements.

Kind regards,

Tungki Pratama Umar, M.D.

Academic Editor

PLOS ONE

Additional Editor Comments (optional):

Reviewers' comments:

Reviewer's Responses to Questions

**Comments to the Author**

1. If the authors have adequately addressed your comments raised in a previous round of review and you feel that this manuscript is now acceptable for publication, you may indicate that here to bypass the “Comments to the Author” section, enter your conflict of interest statement in the “Confidential to Editor” section, and submit your "Accept" recommendation.

Reviewer #1: All comments have been addressed

Reviewer #2: All comments have been addressed

Reviewer #4: All comments have been addressed

2. Is the manuscript technically sound, and do the data support the conclusions?

Reviewer #1: Yes

Reviewer #2: Yes

Reviewer #4: Yes

3. Has the statistical analysis been performed appropriately and rigorously? 

Reviewer #1: Yes

Reviewer #2: Yes

Reviewer #4: Yes

4. Have the authors made all data underlying the findings in their manuscript fully available?

Reviewer #1: Yes

Reviewer #2: (No Response)

Reviewer #4: Yes

5. Is the manuscript presented in an intelligible fashion and written in standard English?

Reviewer #1: Yes

Reviewer #2: Yes

Reviewer #4: Yes

6. Review Comments to the Author

Reviewer #1: (No Response)

Reviewer #2: (No Response)

Reviewer #4: All my previous comments have been fully addressed. There is no additional comment from me and I recommend acceptance of this paper.

7. PLOS authors have the option to publish the peer review history of their article (what does this mean?). If published, this will include your full peer review and any attached files.

Reviewer #1: No

Reviewer #2: No

Reviewer #4: No

---

## [Editor Report · Acceptance letter]

27 May 2024

PONE-D-23-26616R1 

PLOS ONE

Dear Dr. Sun, 

I'm pleased to inform you that your manuscript has been deemed suitable for publication in PLOS ONE. Congratulations! Your manuscript is now being handed over to our production team.

Kind regards, 

on behalf of

Dr. Tungki Pratama Umar 

Academic Editor

PLOS ONE